# Fundamental equations linking methylation dynamics to maximum lifespan in mammals

Steve Horvath [1,2,3] ✉, Joshua Zhang [1], Amin Haghani[1,3], Ake T. Lu[1,3] & Zhe Fei[4] ✉

We describe a framework that addresses concern that the rate of change in any aging biomarker displays a trivial inverse relation with maximum lifespan. We apply this framework to methylation data from the Mammalian Methylation Consortium. We study the relationship of lifespan with the average rate of change in methylation (AROCM) from two datasets: one with 90 dog breeds and the other with 125 mammalian species. After examining 54 chromatin states, we conclude three key findings: First, a reciprocal relationship exists between the AROCM in bivalent promoter regions and maximum mammalian lifespan: AROCM ∝ 1/MaxLifespan. Second, the correlation between average methylation and age bears no relation to maximum lifespan, Cor(Methyl,Age) ⊥ MaxLifespan. Third, the rate of methylation change in young animals is related to that in old animals: Young animals' AROCM ∝ Old AROCM. These findings critically hinge on the chromatin context, as different results emerge in other chromatin contexts.

A fundamental question in biology is why closely related species, such as mammals, exhibit significant differences in maximum lifespans (also referred to as maximum longevity or simply longevity). Years of research have outlined the ecological traits associated with maximum lifespan. In short, maximum lifespan is closely linked to an organism's ability to avoid predation, such as having a large body size, the capacity to fly, or the skill to burrow underground[1–7]. The rate of living theory suggests that the faster an organism's metabolism, the shorter its lifespan. This theory was initially proposed by Max Rubner in 1908, following his observation that larger animals lived longer than smaller ones, and that these larger animals had slower metabolisms. Rubner's rate of living theory, which was once accepted, has now largely been debunked. This shift in perception is due to the application of modern statistical methods which account for the effects of both body size and phylogeny. When these factors are appropriately adjusted for, there is no correlation between metabolic rate and lifespan in mammals or birds[3,7]. Nonetheless, contemporary adaptations of the original rate of living theory have emerged. Many articles have explored the relationship between the rate of change in various molecular markers and maximum lifespan. Specifically, maximum lifespan has been connected to the rates of

change in telomere attrition[8–13], somatic mutations[14–18], and cytosine methylation[19–24].

The strong correlation between maximum lifespan and the rate of change in various factors (telomeres, somatic mutations, methylation) raises the possibility that these relationships might merely be artifacts resulting from the definition of the rate of change. In other words, the strong correlation with maximum lifespan could be a mathematical consequence stemming from the calculation of rates of change per year. To address this concern, we introduce a framework that links the rate of change in a biomarker to maximum lifespan. We show that any biomarker of similar positive correlations with age across multiple species will exhibit an inverse relation with maximum lifespan. Thus considerable care must be taken to avoid biases deriving from the definition of the rate of change.

## Results

We examine the correlation between methylation dynamics throughout the lifespan and maximum lifespan using two datasets from the Mammalian Methylation Consortium[25]. The first dataset encompasses blood methylation data from 90 distinct dog breeds, while the second dataset encompasses many different tissue types from 125 mammalian

[1]Department of Human Genetics, University of California, Los Angeles, CA, USA. [2]Department of Biostatistics, University of California, Los Angeles, CA, USA. [3]Altos Labs, San Diego, CA, USA. [4]Department of Statistics, University of California, Riverside, CA, USA. ✉e-mail: shorvath@altoslabs.com; zhe.fei@ucr.edu

species. We begin with the simpler dog dataset, which consists of a single tissue, to familiarize the reader with our methodology. Our primary scientific emphasis is on the mammalian lifespan dataset, which we analyze in depth.

## Rate of change in methylation in dog breeds

Dog breeds exhibit a striking range of lifespans, with some breeds living up to twice as long as others. A prior study involving two dog breeds indicated an inverse relationship between methylation change rate and breed-specific lifespan[19]. Here we examine the connection between methylation change rate and the lifespans of dog breeds using a substantially larger dataset: $N = 742$ distinct blood samples from 90 different dog breeds[26]. For later applications, we will use the more abstract term stratum to replace the dog breed. The Average Rate of Change in Methylation (AROCM) quantifies the velocity or gradient of age-related changes in a set of cytosines across samples from a given stratum. Therefore AROCMs depend on the age interval and the set of cytosines. Assuming there are $n$ (blood) samples within one stratum (dog breed) and an age range $[\ell, u]$, the methylation matrix is

$$M_{n \times m} = (CG_{ij}), \qquad i = 1, 2, .., n; \; j = 1, 2, .., m,$$

where rows represent (blood) samples and columns represent a set of $m$ CpGs (e.g., CpGs located in a specific chromatin state). For the $i$-th sample, the average methylation value Methyl is defined as:

$$\text{Methyl}_i = \frac{1}{m} \sum_{j=1}^{m} CG_{ij}. \tag{1}$$

To enable comparisons with other aging biomarkers, we define the scaled methylation value as:

$$\text{ScaledM}_i = \frac{\text{Methyl}_i - \text{Mean}(\text{Methyl}_{[1:n]})}{\text{SD}(\text{Methyl}_{[1:n]})}, \tag{2}$$

where the mean and the standard deviation are taken over samples $i = 1, 2, \ldots, n$ within each stratum. Now we define the AROCM in one stratum as the coefficient estimate $\beta_1$ resulting from the univariate linear regression model:

$$\text{ScaledM}_i = \beta_0 + \beta_1 Age_i. \tag{3}$$

The term average in AROCM reflects that Methyl was defined as the average value across a specific set of cytosines (equation (1)). In our analysis of the dog data, we concentrated on the 552 CpGs situated within the BivProm2+ chromatin state, also known as the Bivalent Promoter 2 state that is associated with the Polycomb Repressive Complex 2 (PRC2). The rationale behind selecting this specific chromatin state will be presented in our subsequent application concerning mammalian maximum lifespan. A comprehensive description of chromatin states can be found in Methods. In Methods (equations (11) to (14)), we derive

$$\text{AROCM} = \beta_1 = \frac{\text{Cor}(\text{Methyl}, \text{Age})}{\text{SD}(R)} \frac{1}{\text{Lifespan}}, \tag{4}$$

where $R \doteq \frac{Age}{Lifespan}$ is the relative age in each stratum. Equation (4) reveals that the inverse association between AROCM and maximum lifespan would hold when the first ratio term $\frac{\text{Cor}(\text{Methyl},\text{Age})}{\text{SD}(R)}$ approximates some constant across the strata (Fig. 1a, b). This raises the concern that the relationship between the rate of change (AROCM) and 1/Lifespan might simply result from selecting CpGs with a positive age correlation. To mitigate this concern, we avoided pre-filtering CpGs based on their age correlations. We chose to define the rate of change in relation to sets of CpGs associated with chromatin states that were defined with respect to histone marks (Methods). In practical data applications, inconsistent age ranges and the associated variability in SD ($R$) notably influence the estimation of AROCM and Cor(Methyl, Age). The dependence of AROCM and Cor(Methyl, Age) on SD ($R$) usually is

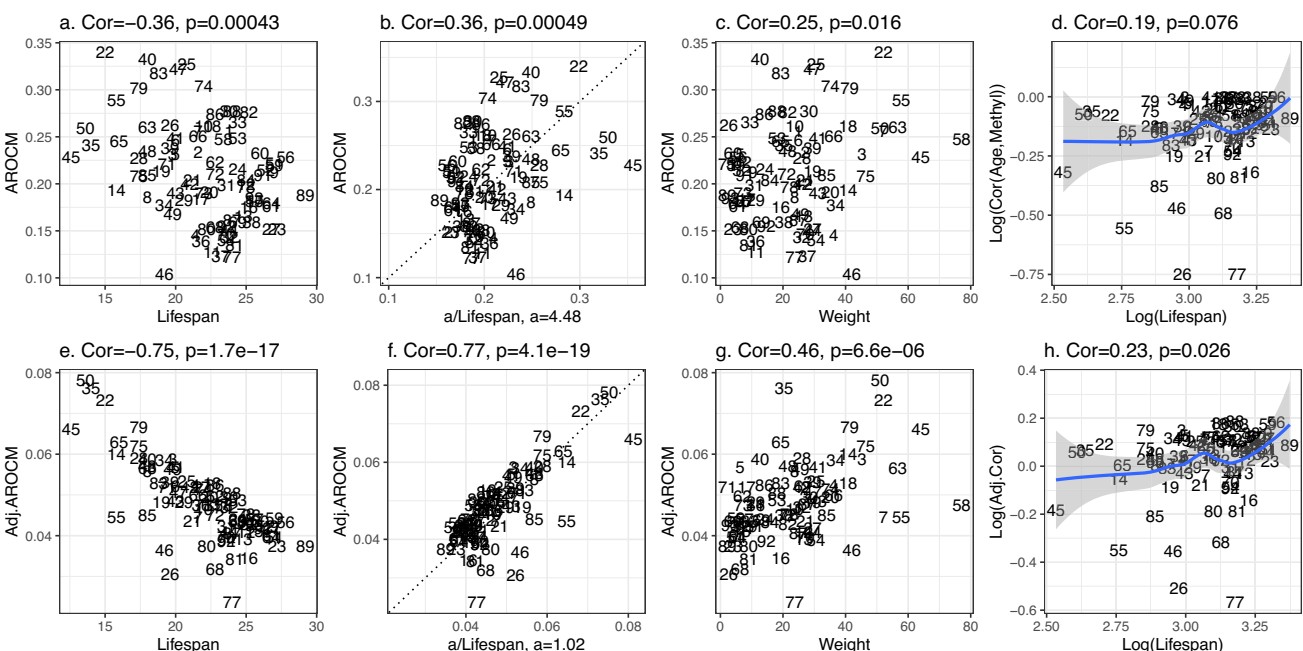

**Fig. 1 | AROCMs in blood samples from $S$ = 90 dog breeds.** Top panels connect AROCMs to **a** Lifespan, **b** 1/Lifespan, **c** adult weight, and **d** Log(Cor(Age, Methyl)) versus Log(Lifespan); bottom panels tie *Adj. AROCM* to the same factors (**e**, **f**, **g**), and **h** Log(Adj.Cor) versus Log(Lifespan). Pearson correlations and corresponding two-sided $t$-test $p$-values are reported in each panel. Gray error bands in **d**, **h** are 95% confidence bands of the LOESS smooth curve fitted to the data. Both adjusted and unadjusted AROCMs were computed with $p = 552$ CpGs in bivalent promoter 2 bound by polycomb repressive complex 2 (PRC2) (BivProm2+)[57]. Each integer label corresponds to a different dog breed (Supplementary Data 1).

not of biological interest. Instead, it indicates flaws in sample selection and study design. To mitigate the impact of these sampling imperfections, we introduce adjusted values:

$$\text{Adj.}AROCM \doteq AROCM \times SD(R)^{1-p}. \tag{5}$$

This straightforward adjustment multiplies AROCM by a power term of $SD(R)$. It offers the additional benefit of establishing a simple relationship with maximum lifespan (Methods equations (17) to (19)):

$$\text{Adj.}AROCM \approx \frac{\text{Adj.}Cor(\text{Methyl, Age})}{\text{Lifespan}},$$
$$\text{where Adj.}Cor(\text{Methyl, Age}) \doteq \frac{Cor(\text{Methyl, Age})}{SD(R)^p}. \tag{6}$$

To arrive at a pronounced inverse correlation between Adj.AROCM and Lifespan, equation (6) suggests minimizing the coefficient of variation in Adj.$Cor$(Methyl, Age) (see equation (27) in Methods for more details). When selecting $p$, we do not take lifespan information into account. However, this criterion for choosing $p$ faces a statistical challenge: overfitting, i.e., resulting in an artificially tight inverse relation between the rate of change and lifespan. Therefore, we advise presenting results for both the original and adjusted values of AROCM in practical applications. Since $p = 0$ implies Adj.$Cor = Cor$, smaller $p$ indicates a lesser degree of sampling imbalance and therefore smaller adjustment. According to the criterion for choosing the adjustment power, $p = 0.1$ emerges as a suitable choice for our dog methylation data (Supplementary Fig. 1).

The low correlation between Adj.$Cor$(Methyl, Age) and breed lifespan on a logarithmic scale ($r = 0.23$, $p = 0.026$, Fig. 1h) implies a strong negative relationship between Adj.AROCM and lifespan on the log scale, as indicated by Proposition 4. This is validated with a correlation coefficient of $r = -0.75$ (Fig. 1e). We also identify positive correlations between AROCM and average breed weight (Fig. 1c, g), which is predictable since breed weight inversely correlates with breed lifespan. Breed characteristics and methylation-based traits like AROCM values for individual breeds are detailed in Supplementary Data 1.

While the adjusted version of AROCM, Adj.AROCM, comes with drawbacks such as the risk of overfitting and reduced interpretation, it bears several benefits. First, it often boosts the data signal ($r = -0.75$ for the adjusted AROCM versus $r = -0.36$ for the unadjusted AROCM, Fig. 1a, e). Second, it aids in deriving formulas for constants of proportionality (equation (25) in Methods). For instance, the constant is 1.02 in equation (25) among dog breeds in our data (Fig. 1f). In summary, both the adjusted and unadjusted AROCMs display negative correlations with lifespan when viewed on a logarithmic scale. Yet, the correlation is notably stronger in the adjusted AROCM when compared to its unadjusted counterpart. This indicates that the adjusted version adeptly counteracts the limitations of an imperfect dataset, particularly the confounding influence of varying $SD(R)$ values.

### Mammalian methylation data and chromatin states
Recent investigations have revealed a connection between the rates of methylation change and the maximum lifespan of mammals[19–21,23,24]. In the present study, we revisit this question utilizing a comprehensive dataset from the Mammalian Methylation Consortium[25,27]. These data are well suited for comparative aging rate studies, as the mammalian array platform provides a high sequencing depth at CpGs, which are highly conserved across mammalian species[28]. We maintained the same definitions as in our dog dataset, though strata were classified by species and tissue types within species. In order to derive reliable AROCM estimates, we removed outlying species/tissue strata using criteria detailed in Methods. The analysis of the AROCM defined with respect to the entire age range [0, Lifespan) involves $S = 229$ distinct

species/tissue strata. We first calculated AROCMs in all species-tissue strata (equation (4)) and aggregated them by species for all 54 chromatin states identified on the mammalian array (Figs. 2; Supplementary Data 2). Specifically, the species-level AROCM is defined as the median across tissue types within that species.

Given that AROCM estimates may exhibit a non-linear relationship with the inverse of maximum lifespan (Fig. 2b), we calculated the Spearman correlation (as opposed to the Pearson correlation) between AROCM and $1/L$, the lifespan. The Spearman correlation coefficients of all chromatin states varied from $-0.44$ (ReprPC4-) to $0.66$ (BivProm1+, BivProm2+) (Fig. 2a and Supplementary Data 2). First, we will discuss chromatin states such as TxEx4-, ReprPC4-, and Quies1-. These states typically exhibit high methylation values[25] and their AROCM estimates frequently result in negative values. Such chromatin states, with frequently negative AROCM estimates, typically exhibit a negative correlation with 1/Lifespan, as indicated by the Spearman correlation in Fig. 2a. For these states, lower negative AROCM values correspond to shorter lifespans. Essentially, species experiencing rapid age-related methylation loss in TxEx4-, ReprPC4-, and Quies1- generally have shorter lifespans. However, the lifespan correlations for chromatin states with negative AROCM values are not as pronounced as those for states with positive AROCM values. We will primarily focus on chromatin states that typically exhibit low methylation values and their AROCM estimates frequently result in positive values. Specifically, the 552 CpGs in the BivProm2+ chromatin state show the strongest positive correlation with 1/Lifespan. This is why we used this chromatin state in our prior application to dog breeds. It is worth noting that similar results can be observed in other chromatin states with low methylation values, such as BivProm1+ and ReprPC1+.

### Non-linear relationship across the life course
In various species-tissue strata, a non-linear relationship exists between age and scaled mean methylation ScaledM (equation (2)), evident in species like the green monkey, pig, and beluga whale (Fig. 2d–f). A similar non-linear pattern is observed for relative age $R$ (Fig. 3). Our Methods section details a parametric model for this relationship, as described in equations (29) and (30). This non-linearity poses a mathematical challenge in estimating AROCM. However, this can be addressed by dividing the age range into segments where linear relationships are appropriate (Fig. 2d–f). Roughly linear relationships can be observed when focusing on either young animals (defined by $R < 0.1$) or older animals ($R \geq 0.1$) (Fig. 3). For instance, in humans, we designated the young stratum by $R < 0.1$, corresponding to $Age < 12.2$ years, while the older stratum was defined as $Age \geq 12.2$. Using the definitions of Young and Old, we computed the AROCM for three specific chromatin states: BivProm2+, BivProm1+, and ReprPC1+. For all three states, we observed two main results: (i) AROCM values are higher in young animals than in older ones; (ii) a pronounced positive correlation exists between AROCM values in young and old animals, with a Pearson correlation coefficient of $r > 0.78$ (see Fig. 4). The strong positive correlation ($r > 0.78$) is noteworthy, even though it can be mathematically derived under specific assumptions, as outlined in Methods, concerning the relationship between AROCM$_{young}$ and AROCM$_{old}$. The resulting biological insight into the connection between aging rates in young and old animals was made possible due to the extensive sample size provided by our Mammalian Methylation Consortium. To address any concerns regarding the 0.1 threshold for $R$, we performed comprehensive sensitivity analyses using thresholds from 0.2 to 0.9, confirming the robustness of our findings (see Supplementary Data 3).

### Adjusted AROCM approximates the inverse of the maximum lifespan
Given that the coefficient of variation for Adj.AROCM is substantially lower than that for Lifespan, our mathematical framework predicts an

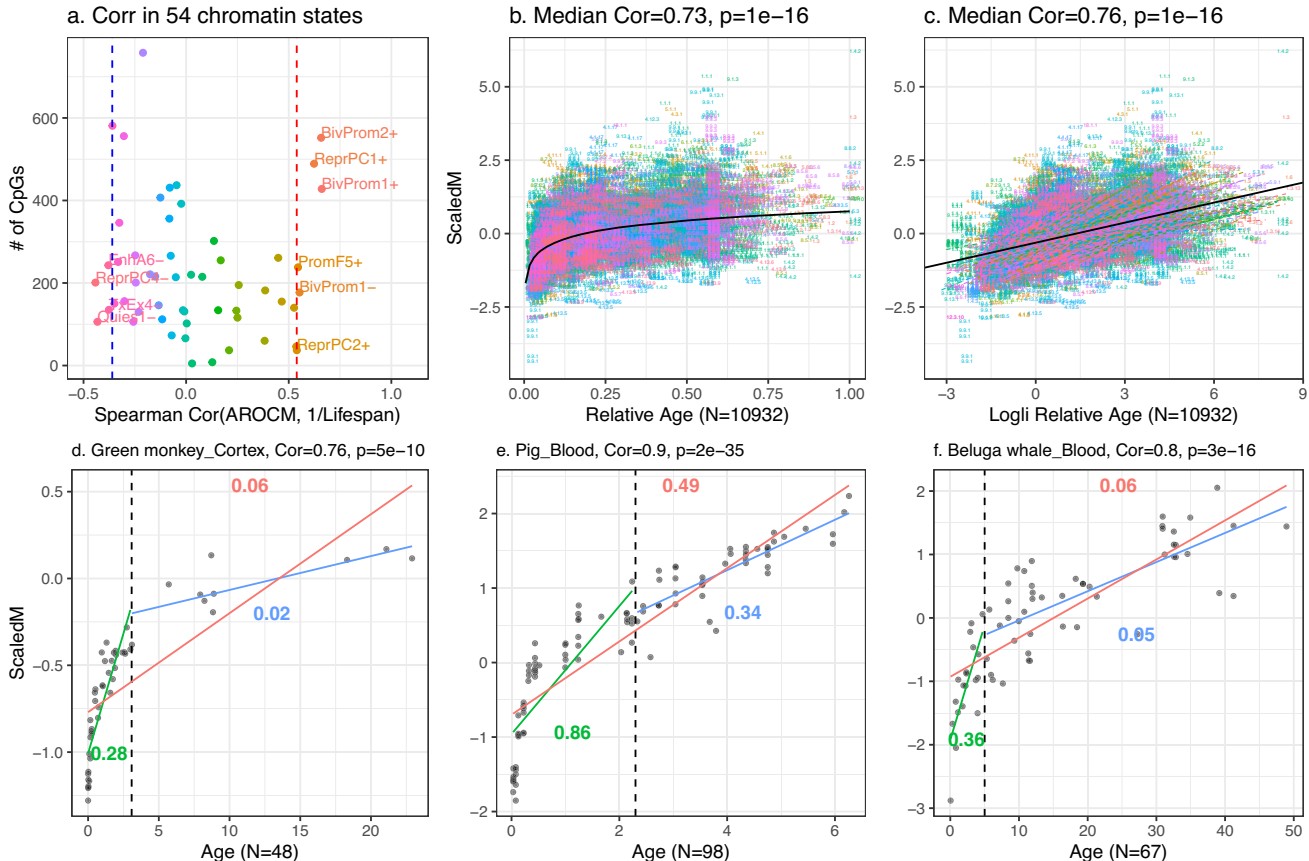

**Fig. 2 | Age versus mean methylation in select chromatin states in mammals.**
**a** Displays the Spearman correlation between the inverse of mammalian maximum lifespan $1/L$ (x-axis) and the AROCM of diverse chromatin states. Each dot represents one of the 54 universal chromatin states[57], with the AROCM defined per Methods. Only chromatin states with Spearman correlation exceeding 0.6 or below −0.3 with the inverse of maximum lifespan are labeled. More details are in Supplementary Data 2. The most pronounced correlation is found for CpGs in bivalent promoter 2 also bound by polycomb repressive complex 2 (BivProm2+). **b** Scaled Mean Methylation versus Relative Age across all samples; **c** Scaled Mean

Methylation against Log-linear Transformed Relative Age in all samples. Black curves mark the overall trend. In panel **c**, a linear regression line was fitted per species. **d–f** Depicts AROCM calculation for BivProm2+ state in 3 species-tissue strata. Colored line segments/numbers in each panel represent: Young AROCM for age interval $[0, 0.1L]$ in green; Old AROCM for age interval $(0.1L, L)$ in blue; AROCM for age interval $[0, L]$ in red. Pearson correlations and corresponding two-sided t-test p-values are reported in **b–f**. The sample size for each panel is reported on the x-axis label. Source data are provided as a Source Data file.

inverse correlation between Adj.AROCM and Lifespan, as indicated by equation (6) and Proposition 4. Indeed, both the AROCM ($r = −0.85$) and the adjusted AROCM ($r = −0.92$) exhibit strong negative correlations with maximum lifespan on the log scale (Fig. 5a, c). The difference between the two measures is more pronounced when looking at the original scale (no log transformation): the adjusted AROCM leads to a high correlation with $a$/Lifespan ($r = 0.96$) compared to that for the unadjusted AROCM ($r = 0.72$, Fig. 5b, d). A benefit of the adjusted correlation Adj.Cor(Methyl, Age) is that its mean value approaches 1.0 across species (mean = 1.11, Supplementary Fig. 2i) since this results in a simple memorable formula: the adjusted AROCM is roughly equal to 1/Lifespan because $a \approx 1$ (equation (6), Fig. 5d). Supplementary Data 4 provides detailed results on AROCMs and their adjusted versions.

Strong negative correlations with lifespan can be further observed when AROCM is defined with respect to young animals only ($R < 0.1$) or old animals only ($R \geq 0.1$, Supplementary Fig. 3). Since maximum lifespan is positively correlated with average age of sexual maturity and gestation time, we find that both variables correlate strongly with AROCM as well (Supplementary Fig. 4). Our findings underscore the efficacy of the adjusted AROCM in addressing the issue of strata with widely varying values of SD ($R$). Traditionally, one might handle the variability in SD ($R$) by limiting the analysis to strata that have

approximately similar SD ($R$) values. To demonstrate that our main findings are consistent using this conventional method, we conducted the same analysis on strata with comparable SD ($R$) values. The results confirmed our earlier findings of strong correlations between AROCM and 1/Lifespan, with correlation coefficients of $r = 0.99$ for strata within SD ($R$) $\in (0.2, 0.4)$ and $r = 0.97$ for strata within SD ($R$) $\in (0.4, 0.6)$, respectively (see Supplementary Fig. 5). Similarly, a stratified analysis reaffirmed our key findings from the dog breed study (refer to Supplementary Fig. 6).

Our outlier removal algorithm (Methods) was crucial for establishing strong relationships between AROCM and maximum mammalian lifespan. Without this algorithm, the relationships are substantially weaker (Fig. 6).

**Methylation-age correlation does not relate to lifespan**
The correlation between Mean Methylation and Age, denoted as Cor(Methyl, Age), assumes similar values for long and short-lived species, e.g., Median Cor(Methyl, Age) = 0.51 for humans, 0.58 for humpback whale, 0.62 for Asian elephant, 0.66 for mouse, 0.67 for brown rat, 0.69 for prairie vole. As shown in Fig. 1d, Cor(Methyl, Age) does not relate to lifespan in the dog data. To delve deeper into the relationship between Cor(Methyl, Age) and maximum lifespan, we grouped each species into one of five groups based on their maximum

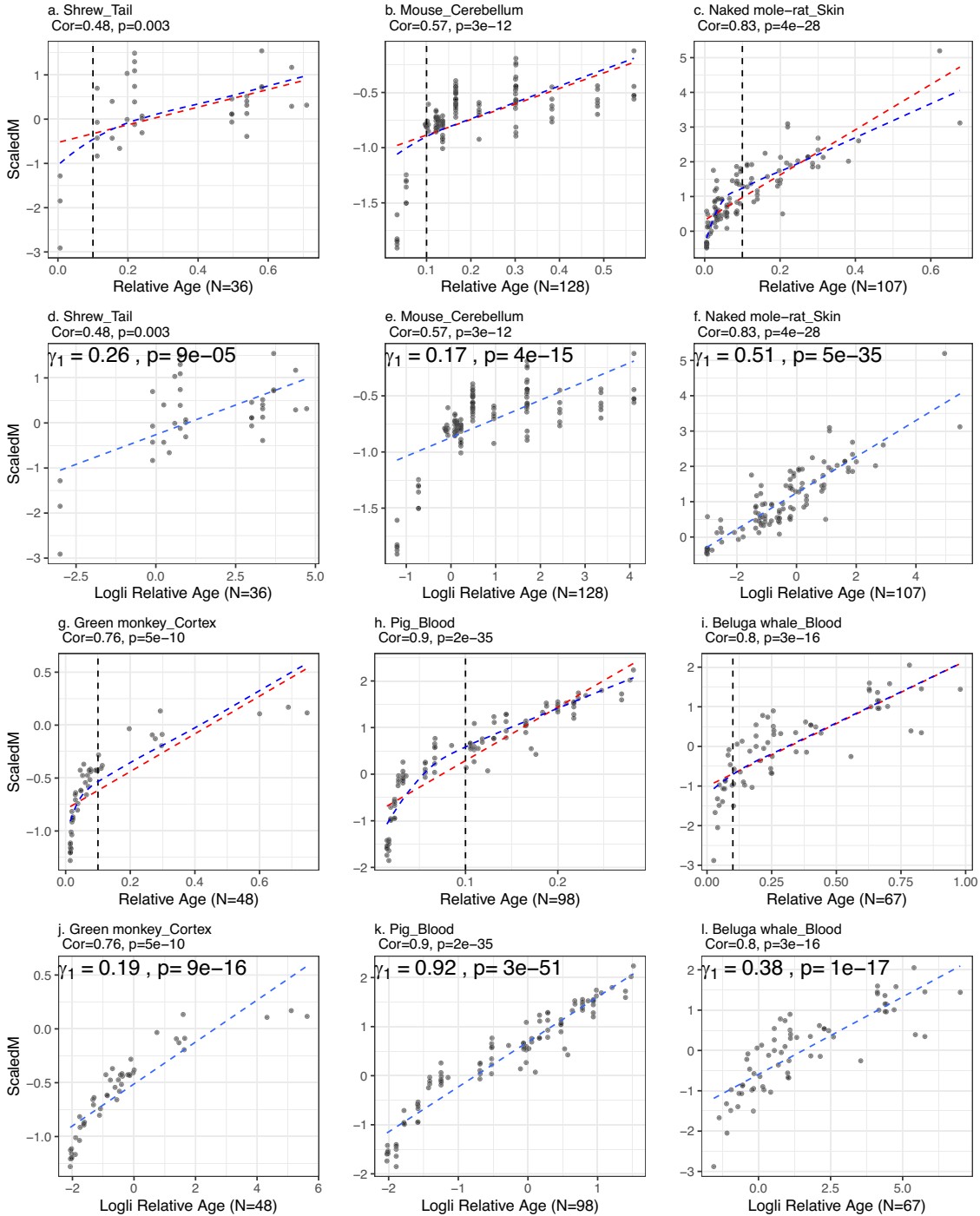

**Fig. 3 | Scaled mean methylation against relative age or log-linear transformed relative age in 6 additional species-tissue strata.** The correlations increase and associations become more linear post-transformation. **a–c**, **g–i** Scaled Mean Methylation versus Relative Age, the red dashed line is the linear fit to relative age, the blue dashed curve represents the inverse transformation of the linear fit in (**d–f**, **j–l**); **d–f**, **j–l** Scaled Mean Methylation versus Log-linear transformed relative age, where $\gamma_1$ is the AROCM defined in equation (29), with $ScaledM_i = \gamma_0 + \gamma_1 T_i$. A two-sided z-test is used to calculate the p-values for $\gamma_1$. Pearson correlations and corresponding two-sided t-test p-values are reported in each panel. The sample size for each panel is reported on the x-axis label. Source data are provided as a Source Data file.

lifespan: less than 10 years, 10–19 years, 20–24 years, 25–39 years, or over 40 years. These groupings were designed to ensure a balanced number of species per group. Our analysis revealed no discernible trend between either Cor(Methyl, Age) or Adj.Cor and the lifespan group (Fig. 7). Similarly, we find no relationship between lifespan and Cor(Methyl, Age) when the latter is defined with respect to young or old animals (Methods). The analysis mentioned above overlooks a technical challenge. In our dataset, Cor(Methyl, Age) shows a

correlation with the standard deviation of relative age, with a Pearson correlation of $r = 0.23$ in all species-tissue strata and $r = 0.27$ across species (Supplementary Fig. 2d, e). This correlation is not ideal, as it stems from an imbalanced and imperfect data sampling. Ideally, with all species sampled to have the same distribution in relative age, SD $(R)$ would remain constant, making Cor(Methyl, Age) and SD $(R)$ independent (Methods). The variability in SD $(R)$ led to the introduction of the adjusted correlation, Adj.Cor (equation (5)). Using an adjustment

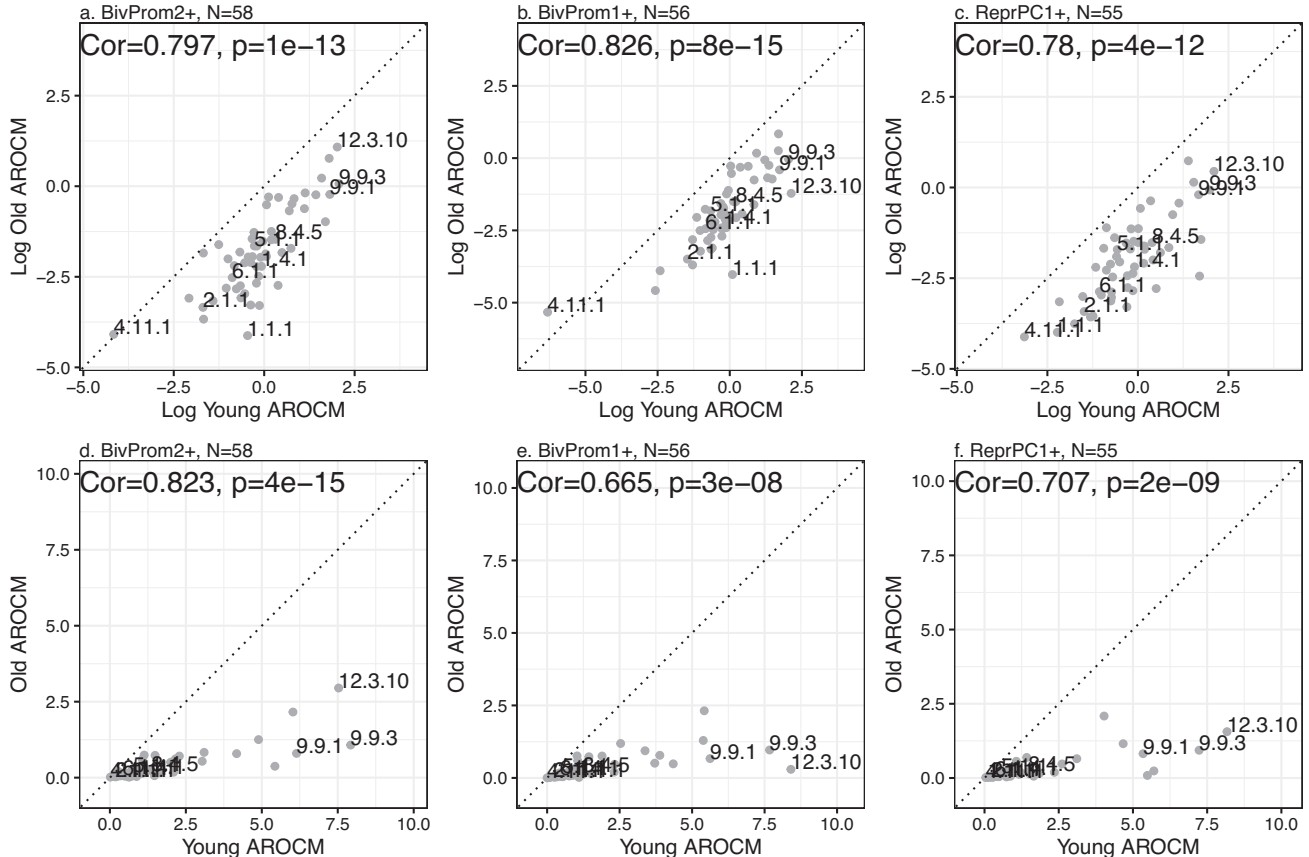

**Fig. 4 | AROCM in young animals versus that in old animals. a, b, c** Natural log-transformed values of AROCM in young animals versus AROCM calculated in old animals. Mean methylation was defined with respect to CpGs located in chromatin state **a** BivProm2+, **b** BivProm1+, **c** ReprPC1+. **d, e, f** Corresponding plots on the original scale, i.e., without log transformation. Each dot corresponds to a different species. Median aggregation was used to combine AROCM results from different tissues by species. *x*-axis: AROCM for young samples (Relative Age $R < 0.1$). *y*-axis: AROCM for old samples ($R >= 0.1$). Samples are labeled by mammalian species numbers. Each panel reports the Pearson correlation value Cor and the corresponding two-sided *t*-test *p*-value. Source data are provided as a Source Data file.

power of $p = 0.25$ (Supplementary Fig. 1b), Adj.Cor is found to have a non-significant correlation with SD ($R$) (Supplementary Fig. 2g, h). A more detailed presentation of these results using bar plots is available in Supplementary Figs. 7, 8, and 9.

### Adult weight does not confound the relationship between AROCM and lifespan

Body size (e.g., measured using average adult body mass or adult weight) is a major determinant of maximum lifespan-with larger animals living, on average, longer than smaller ones[7,29–34]. Adult weight is often a confounding factor in comparative studies of maximum lifespan[33,35,36]. To address this concern, we carried out several analyses to demonstrate that adult weight does not explain the observed relationships between AROCM and maximum lifespan. First, we demonstrate that the correlation between log(AROCM) and log(Weight) is much weaker than that observed for maximum lifespan ($r = -0.66$ for average weight, Supplementary Fig. 10a, versus $r = -0.85$ for maximum lifespan, Fig. 5a). Second, we fit the following multivariate regression models between the unadjusted AROCM, maximum lifespan and adult weight (Table 1),

$$M1 : \log(L) \sim \log(AROCM) + \log(Weight),$$
$$M2 : \log(AROCM) \sim \log(L) + \log(Weight). \quad (7)$$

When lifespan is regressed on AROCM and weight on the log scale, both AROCM and weight are highly significant (model M1, Table 1).

When AROCM is regressed on lifespan and weight on the log scale, lifespan remains highly significant while weight is not (model M2, Table 1). These results demonstrate that adult weight has only a weak effect on the relationship between lifespan and AROCM.

### Effect of tissue type on the relationship between AROCM and lifespan

Most proliferating tissue types have similar values of Cor(Methyl, Age) but slightly lower correlations can be observed for non-proliferating tissues such as the cerebellum, cerebral cortex, skeletal muscle, and heart when looking at all samples irrespective of age (Supplementary Fig. 11). The lowest age correlation values are observed for the kidney, but this finding requires further replication. To carry out a formal analysis of the effect of tissue type, we added indicator variables for tissue types as covariates to the multivariate regression models that explored the relationship between AROCM and lifespan:

$$M3 : \log(AROCM) \sim \log(L) + \log(Weight) + Brain,$$
$$M4 : \log(AROCM) \sim \log(L) + \log(Weight) \quad (8)$$
$$+ Brain + Skin + Liver + Muscle + Tail,$$

where model M4 added indicator variables for several tissue types. Both M3 and M4 allow insights into the potential tissue differences in AROCM estimates, as detailed estimates can be found in Table 1. After including log($L$) as the covariate, only liver tissue had a significant (positive) association with log($AROCM$), ($p = 0.03$).

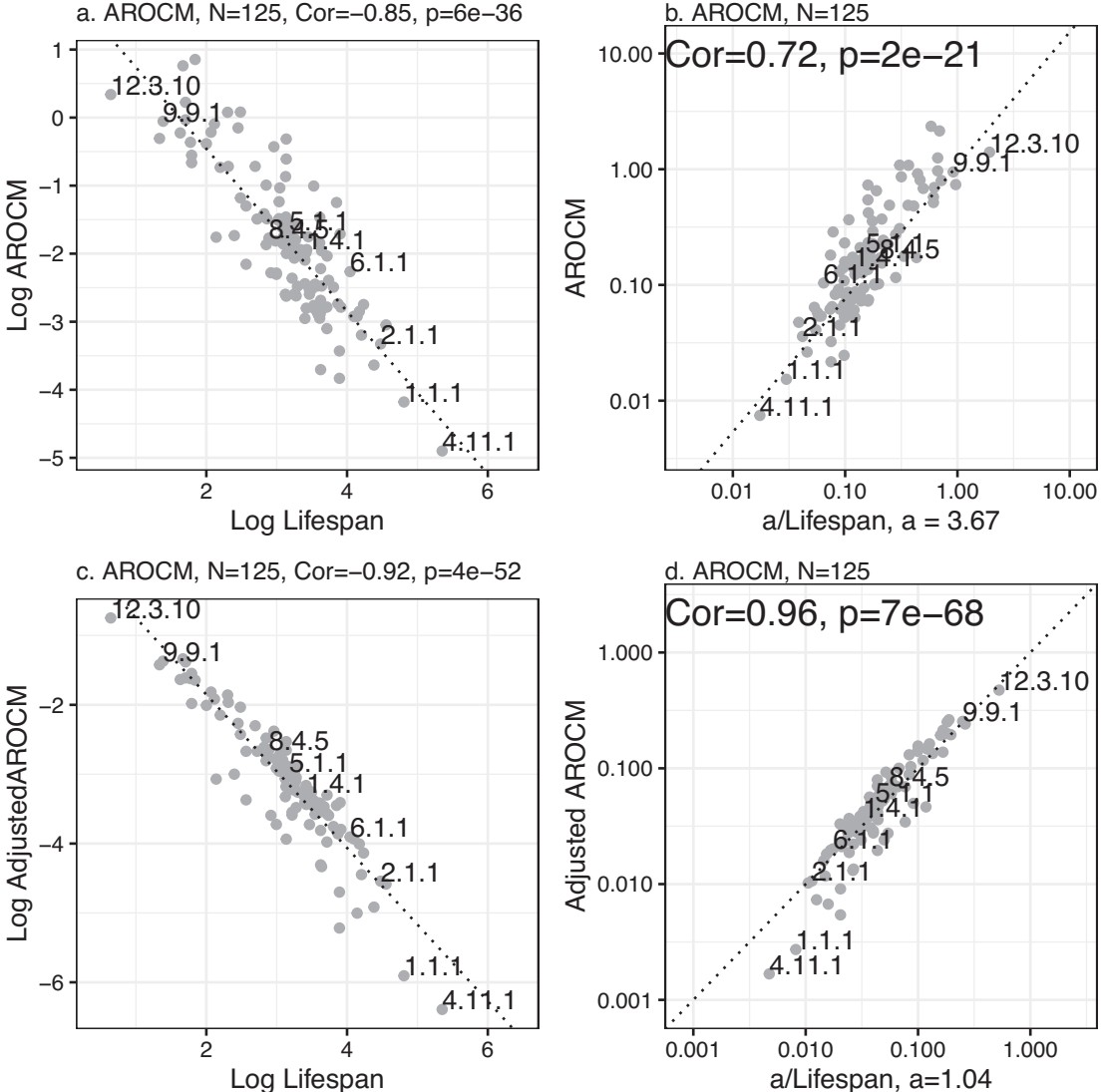

**Fig. 5 | AROCM versus maximum mammalian lifespan excluding outlying species. a** Log AROCM versus Log Lifespan; **b** AROCM against $a$/Lifespan; **c** Log Adj.AROCM versus Log Lifespan (equation (20)); **d** Adj.AROCM against $a$/Lifespan (equation (25)). Each dot reports a species-level result (median across tissues) with $N = 125$ species. Pearson correlations and the corresponding two-sided $t$-test $p$- values are reported for each panel. Dotted lines in **b**, **d** denote $x = y$. Supplementary Fig. 3 contains results for young and old AROCM. We conducted an analogous analysis on strata with comparable SD ($R$) values, see Supplementary Fig. 5. Source data are provided as a Source Data file.

## Phylogenetic effects

The shared genetic, behavioral, and ecological traits among related species, inherited from common ancestors, can create non-random associations or phylogenetic signals. Ignoring these phylogenetic relationships can lead to false correlations or overlook real ones[33,36,37]. Upon adjusting for phylogenetic relationships, the previously mentioned associations persist, albeit with weaker correlation coefficients. In the case of the dog methylation data, we observed substantial positive associations in the phylogenetically independent contrasts (PICs[37]) of AROCM and breed lifespan (refer to Supplementary Fig. 12). Here, we recorded a correlation of $-0.26$ ($p = 0.018$) for AROCM, and a correlation of $-0.64$ ($p = 7.3 \times 10^{-11}$) for Adjusted AROCM.

In the mammalian data, we again found significant positive associations in PICs of AROCM and maximum lifespan (Supplementary Fig. 13). Correlations for this dataset were $r = -0.2$ ($p = 0.03$) for AROCM, $-0.35$ ($p = 0.00013$) for Adjusted AROCM, and $-0.23$ ($p = 0.022$) for Adjusted Old AROCM. Collectively, these findings

indicate a connection between the rate of methylation change in bivalent promoter regions and maximum lifespan, which holds even after phylogeny adjustment.

## Discussion

We established the Mammalian Methylation Consortium with two primary objectives. The first objective was to develop DNA methylation-based measures to track the passage of time, culminating in the creation of the pan-mammalian methylation clock[27]. The second objective aimed to understand the epigenetic correlates of maximum mammalian lifespan, a goal explored through a trilogy of papers. In the first paper, we developed multivariate predictors of maximum lifespan based on cytosine methylation[38]. This predictor can estimate species-specific characteristics, such as maximum lifespan, from a DNA sample, even if the species is unknown. The second paper characterized individual CpGs and clusters of CpGs (modules) that correlate with maximum lifespan[25], providing insights into the methylation landscape

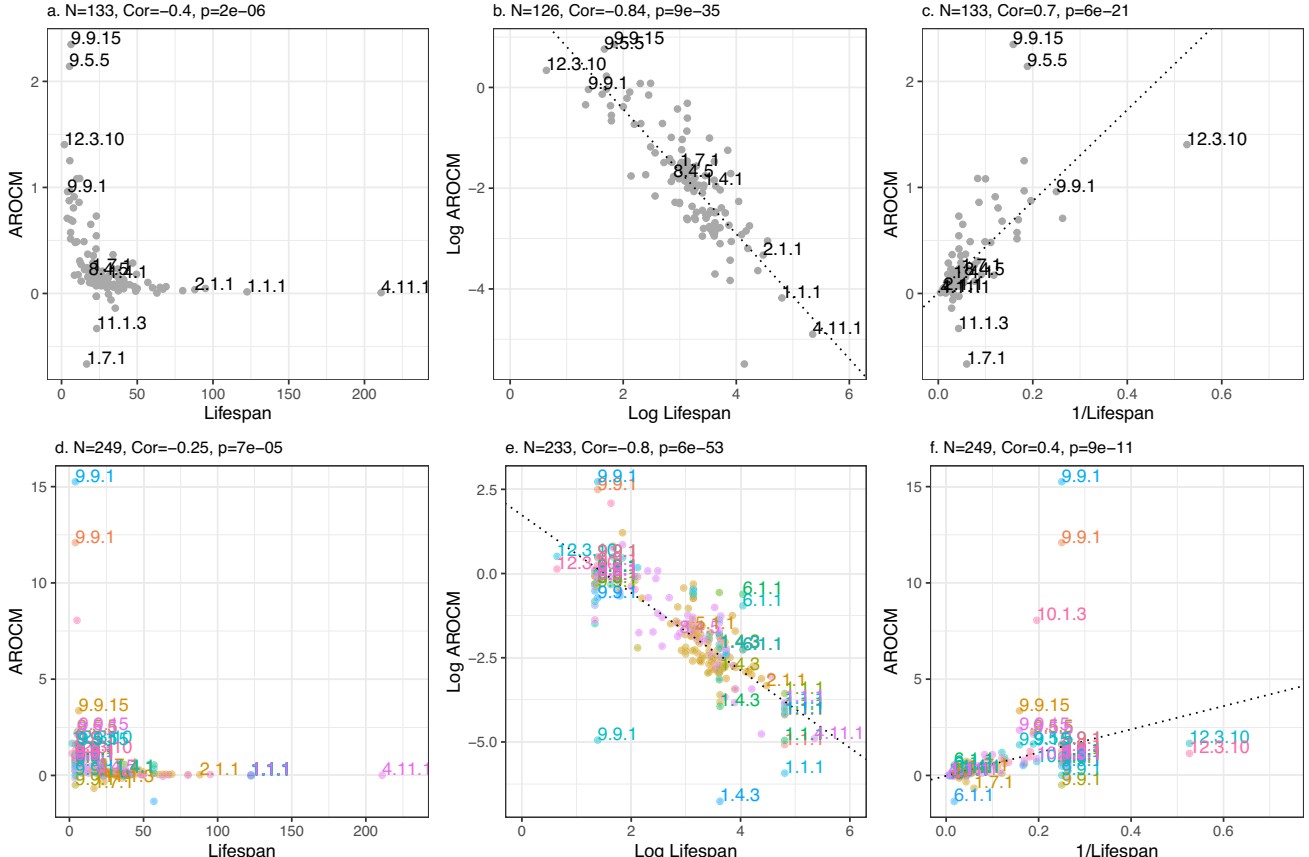

**Fig. 6 | AROCM versus maximum mammalian lifespan including outlying species.** This figure is similar to Figure 5 but it includes the outlying species and strata, i.e., we skipped the filtering approach described in Methods. **a**–**c** Each dot reports a species-level result (median across tissues) with *N* = 133 species. **d**–**f** Species-tissue strata analysis based on *N* = 249 species-tissue strata (represented by the dots). **a**, **d** AROCM versus Lifespan; **b**, **e** Log AROCM versus Log Lifespan; **c**, **f** AROCM against 1/Lifespan; Pearson correlations and the corresponding two-sided *t*-test *p*-values are reported for each panel. Source data are provided as a Source Data file.

of long-lived species. Interestingly, this study found only a weak overlap between CpGs that correlate with maximum lifespan and those associated with chronological age. While we understand that a species characteristic like maximum lifespan is genetically hardwired and does not change with the individual's age, this finding challenges the intuitive hypothesis that individual aging (the passage of time) must relate to maximum lifespan (a species characteristic).

The current paper, the final installment in our trilogy on maximum lifespan, addresses this hypothesis by integrating our first objective (correlates of the passage of time, i.e., the rate of change) with our second objective (maximum lifespan). We initially presented these results during a TEDxBerkeley talk in February 2020[39]. It took five years to publish these findings because we discovered that studies linking the rate of change in an aging biomarker to maximum lifespan are inherently biased. Permutation studies revealed that a strong relationship between the rate of change and maximum lifespan could arise even without an actual signal in the data, due to the underlying mathematical definitions. We deemed it necessary to collect a large dataset and to develop a mathematical framework to highlight and characterize this bias, as it holds significant relevance for other aging biomarkers beyond methylation.

### The rate of change is not always inversely proportional to the lifespan

Prior studies have linked the rate of change in methylation to maximum lifespan in smaller samples of mammalian species[19,20,23,40], and

have suggested a strong positive correlation between methylation rate changes and the inverse of mammalian lifespan. But this is not always the case. By leveraging the large dataset from our Mammalian Methylation Consortium and a careful mathematical framework we demonstrate that the strong positive correlation between AROCM and 1/Lifespan is only found in certain chromatin states such as bivalent promoters (Fig. 2a). In other chromatin states, the correlation between AROCM and 1/Lifespan is either not present or even reversed (Supplementary Fig. 14). Future studies might delve deeper into which chromatin regions result in the opposite interpretation, where a rapid rate of change correlates with an increased maximum lifespan. Our research initiated this exploration by pinpointing chromatin states where the rate of change negatively correlates with 1/Lifespan. One intriguing aspect is understanding why chromatin states like TxEx4-, ReprPC4-, and Quies1, which negatively correlate with 1/Lifespan, exhibit a more subdued *p*-value association compared to their positively correlated counterparts. This might be attributed to the varied age-related methylation loss patterns across different tissue types. For example, the TxEx4 high-expression transcription state indicates age-related methylation loss in non-proliferative tissues[27]. Yet, in proliferative tissues such as blood or skin, TxEx4 has diminished enrichment of age-related cytosines undergoing methylation loss[27]. Further, the correlation between mammalian lifespan and AROCM is not significant when employing all 8970 CpGs that correspond to both eutherians and marsupials (Supplementary Fig. 15). These findings collectively underscore that

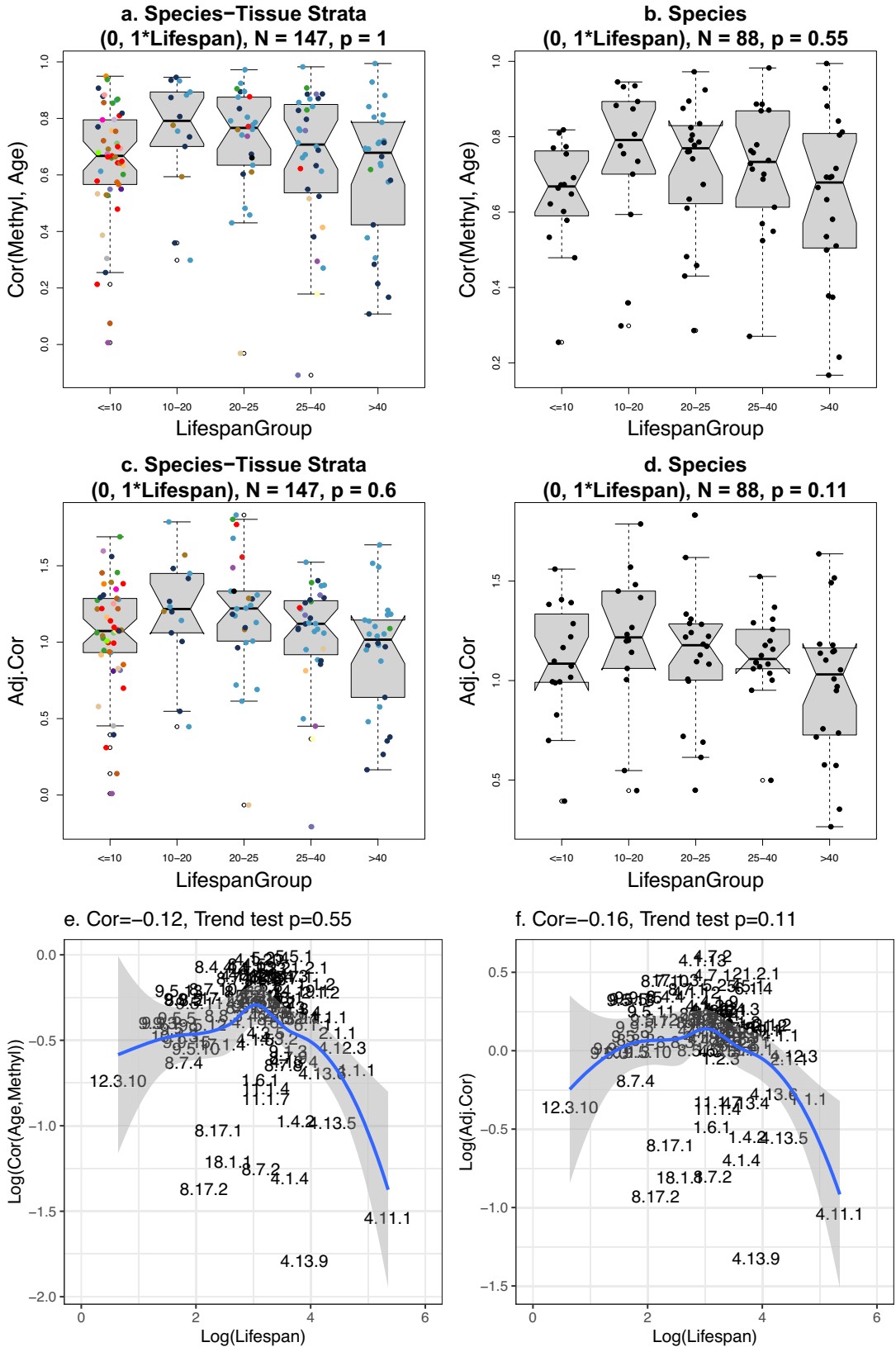

**Fig. 7 | Lifespan group analysis of age correlations.** Cor(Methyl, Age) and Adj.Cor versus groups of species by maximum lifespan. **a**, **b** Non-significant associations between Cor(Methyl, RelAge) and lifespan groups. **c**, **d** Adjusted Cor (equation (18)) by Lifespan groups. **e**, **f** Log of Cor(Methyl,Age) and Adj.Cor versus Log of Lifespan. *p*-values are based on the Mann-Kendall Trend Test. These results illustrate

condition (C1) in Methods. For each box, the dots correspond to individual strata, the center line is the median; the box limits are upper and lower quartiles; the whiskers are 1.5 times the interquartile range. Source data are provided as a Source Data file. Gray error bands in **e** and **f** are 95% confidence bands of the LOESS curve fitted to the data.

**Table 1 | Regression models involving maximum lifespan**

| Model | Covariate | Estimate | Std. error | t-value | Pr( >\|t\|) |
|---|---|---|---|---|---|
| M1: log(Life-span) ~ | (Intercept) | 1.54 | 0.071 | 21.7 | 5.40E-52 |
| | log(AROCM) | −0.44 | 0.026 | −16.92 | 1.80E-35 |
| | log(AverageWeight) | 0.09 | 0.01 | 9.23 | 5.10E-18 |
| M2: log(AR-OCM) ~ | (Intercept) | 1.92 | 0.18 | 10.94 | 9.80E-17 |
| | log(Lifespan) | −1.3 | 0.077 | −16.92 | 1.80E-35 |
| | log(AverageWeight) | 0.015 | 0.02 | 0.76 | 0.45 |
| M3: log(AR-OCM) ~ | (Intercept) | 1.7 | 0.19 | 9 | 1.20E-16 |
| | log(Lifespan) | −1.2 | 0.079 | −15 | 1.80E-35 |
| | log(AverageWeight) | 0.015 | 0.02 | 0.78 | 0.43 |
| | Brain | −0.26 | 0.19 | −1.4 | 0.16 |
| M4: log(AR-OCM) ~ | (Intercept) | 1.79 | 0.22 | 8.24 | 1.94e-14 |
| | log(Lifespan) | −1.28 | 0.08 | −16.06 | 2e-16 |
| | log(AverageWeight) | 0.03 | 0.02 | 1.22 | 0.23 |
| | Brain | 0.03 | 0.21 | 0.16 | 0.87 |
| | Blood | 0.16 | 0.15 | 1.02 | 0.31 |
| | Skin | 0.11 | 0.16 | 0.71 | 0.48 |
| | Liver | 0.50 | 0.23 | 2.13 | 0.03 |
| | Muscle | −0.52 | 0.37 | −1.42 | 0.16 |
| | Tail | 0.19 | 0.33 | 0.59 | 0.56 |

We considered animals across the entire age range. We used mean methylation in BivProm2+ for all species-tissue strata ($N$ = 229). Source data are provided as a Source Data file.

the chromatin context substantially affects the relationship between AROCM and lifespan.

## Chromatin state BivProm2 is special

Bivalent chromatin states generally exhibit low methylation levels in most tissues and species and are bound by polycomb repressive complex 2 (PRC2)[27,41,42]. Many previous articles, including our pan-mammalian aging studies, have shown target sites of PRC2 gain methylation with age in most species and tissues[27,41–43]. We proposed a simple and fundamental equation that links the adjusted rate of change in bivalent chromatin regions to the inverse of maximum lifespan (equation (6)). Note that equation (6) does not involve hidden parameters. Our study strongly suggests that careful definition and measurement of both *Adj. AROCM*$^{(s)}$ and $L^{(s)}$, the choice of power $p$, and the interval of relative age range will result in an actual equality:

$$Adj.AROCM^{(s)} = \frac{1}{L^{(s)}}. \qquad (9)$$

## Invariants

Our mathematical framework demonstrates that equation (9) is a consequence of another major finding from our study: the Pearson correlation between methylation levels and age in selected chromatin states, Cor(Methyl, Age), does not exhibit a strong correlation with maximum lifespan. Given the substantial differences in age correlation values, the term "life history invariant" would be misplaced. However, our studies indicate that age correlation does not relate to maximum

lifespan across mammals (Fig. 7). Thus, the age correlation in short-lived species is similar to that in long-lived species, likely explaining the similar accuracy of epigenetic clocks for both short- and long-lived species[27].

## Development versus aging

Previous research has shown that rates of developmental change in methylation exceed those observed post-puberty in humans[44,45]. Our extensive mammalian dataset supports this observation across a broad range of species, as shown in Fig. 4. Notably, we have established a proportional relationship between the rates of change in young and old animals, a finding consistent across species (Fig. 4). The underlying principle of this proportionality is mathematically articulated in equation (28), which can be derived from the biological premise that a uniform epigenetic maintenance mechanism regulates methylation levels across the lifespan of a species. More specifically, this proportionality emerges from the hypothesis of a continuous, although potentially non-linear, increase in cytosine methylation at specific genomic locations (bivalent promoters) from development through old age. This has been explicitly formulated in our life course equations (29) and (30), assuming a non-linear relationship between scaled methylation (ScaledM) and relative age. From this standpoint, if we consider that identical epigenetic maintenance processes govern methylation increases in both young and old animals, it logically follows that AROCM$_{young}$ and AROCM$_{old}$ would be correlated across different species. Conversely, a lack of correlation would imply fundamentally distinct epigenetic regulation mechanisms at different life stages. Biologically, the proportionality between AROCM$_{young}$ and AROCM$_{old}$ lends support to deterministic aging theories, which propose a connection between developmental changes and those that occur later in life[46–50]. Our theoretical model explains, under the premise that young and old animals share the same epigenetic maintenance mechanisms, why AROCM$_{young}$ and AROCM$_{old}$ are linked by a multiplicative constant, rather than a different type of non-linear but monotonic relationship. Future iterations of our life course equations (29) and (30) may need to be refined to account for a possible early developmental phase characterized by rejuvenation processes[51].

## Limitations and future research

Our research has multiple limitations, which stem from the inherent challenges in reliably calculating a rate of change. Careful examination of outliers in all strata is essential to reliable rate of change estimates, which we only ensured for the top chromatin states. To tackle these issues, we developed an outlier removal algorithm. Without this algorithm, the empirical data aligns far less with the mathematical formulas (Fig. 6). A notable limitation in our study is the wide fluctuation in age ranges across various strata (leading to different values of $SD(R)$). This can significantly influence the estimates of the rate of change. To address this, we introduced adjusted estimates for the rate of change. However, these adjustments are not without issues, including the risk of overfitting and challenges in interpretation. We recommend presenting both the unadjusted and adjusted rate of change estimates in findings, as done in our research. Our results remain consistent whether using adjusted or unadjusted estimates. Other limitations relate to potential confounders such as body size, tissue type, and phylogenetic relationships. Our multivariate model analysis partially addresses these concerns. Our results indicate that the inverse relationship between AROCM and mammalian lifespan persists even after adjusting for these confounders. A promising avenue for research is examining the disparities in methylation rate changes between sexes. Additionally, it would be fascinating to investigate potential interventions influencing the AROCM. Many

interventions influencing the lifespan of mice have been linked to alterations in age-associated methylation[21,51–55].

## Methods

### Ethics

We utilized publicly available data as described in ref. 25. Below, we detail the species involved, institutions, and relevant ethics protocol numbers. Mammalian samples, including yellow-bellied marmots, were collected under the UCLA Institutional Animal Care and Use protocol (#2001-191-01, renewed annually) and with permission from Colorado Parks and Wildlife (TR917, renewed annually). Plains zebra samples were collected under a protocol approved by the Research Safety and Animal Welfare Administration, University of California, Los Angeles: ARC #2009-090-31, originally approved in 2009. General mammalian samples were approved by the Animal Welfare and Ethics Review Board, University of Rochester Committee on Animal Resources (UCAR), Animal Protocol #101939/UCAR-2017-033. Human samples were covered by the University of California, Los Angeles (IRB#15-001454, IRB#16-000471, IRB#18-000315, IRB#16-002028) and the Oxford Research Ethics Committee in the UK (reference 10/H0605/1). Voles were managed under the Institutional Animal Care and Use Committee (IACUC) of Cornell University (protocol #2013-0102), following NIH guidelines. Deer mice were handled by the Peromyscus Genetic Stock Center, University of South Carolina, with approval from the IACUC of the University of South Carolina (protocol #2356-101506-042720). Horses were covered under the UC Davis IACUC protocols (#19037, #20751, and #21455). The naked mole rat study was approved by the University of Rochester Committee on Animal Resources (protocol #2009-054). Beluga whale research was authorized under NMFS Research Permit 932-1905-00/MA-009526 and MMPA Permit #20465, issued by the National Marine Fisheries Service (NOAA). Bowhead whale studies were approved by Fisheries and Oceans Canada (DFO), LFSP S-19/20-1007-NU, and Animal Care approval (AUP) FWI-ACC-2019-14. Killer whale research was conducted under NMFS General Authorization No. 781-1725 and scientific research permits 781-1824-01, 16163, 532-1822-00, 532-1822, 10045, 18786-03, 545-1488, 545-1761, and 15616. Humpback whale research was approved under various permits, including NMFS permits (21485, 16325, 20465, 14245, 633-1483, 633-1778, 932-1905), the Canadian Department of Fisheries and Oceans, and IACUC #NWAK-18-02. For cats, the research was approved by the Clinical Research Ethical Review Board of the Royal Veterinary College (URN: 2019 1947-2). The elephant study was authorized by the management of each participating zoo, reviewed by zoo research committees where applicable, received IACUC approval (#18-29) at the National Zoological Park (Smithsonian's National Zoo), and was endorsed by the Elephant Taxon Advisory Group and Species Survival Plan. Rat studies were approved by the Institutional Animal Ethics Committee of SVKM's NMIMS University, Mumbai (approval no. CPCSEA/IAEC/P-6/2018), adhering to CPCSEA guidelines from the Government of India. Dog research was approved by the Animal Care and Use Committee of the National Human Genome Research Institute (NHGRI) at NIH (protocol #8329254). Bat research was conducted with approval from the University of Maryland IACUC (protocol FR-APR-18-16). Cattle research was approved by the University of Nebraska IACUC (approval #1560). Mice research was approved by the University of Texas Southwestern Medical Center (APN 2015-100925, renewed every 3 years). Apodemus mouse research was approved by the University of Edinburgh Ethical Review Committee (UK Home Office Project License PP4913586). Spiny mouse research was conducted under the approval of the University of Kentucky (protocol #2019-3254). Finally, shrews and other small species from the Museum of Biological Diversity at The Ohio State University were managed under The Ohio State University IACUC (protocol #2017A00000036).

### Statistics and reproducibility

We do not present findings from specific experiments. Rather, DNA samples were collected opportunistically from available freezer-stored materials provided by our collaborators. Data collection and analysis were not conducted blind to variables such as tissue type. Methylation measurements were taken from different animals, ensuring that no animal was measured more than once. Our linear regression modeling and correlation tests assume normality. Severe outliers were excluded to arrive at a reliable rate of change estimates. To this end, we developed an outlier removal algorithm described below. Without it, the empirical data aligns poorly with the mathematical formulas (see Fig. 6). Below, we outline the quality control measures for our samples and the statistical methods used in each analysis. Additional details are provided in Supplementary Note 1.

### Selection of dog methylation data

We analyzed methylation profiles from $N = 742$ blood samples derived from 93 dog breeds (Canis lupus familiaris). Primary characteristics (sex, age, average life expectancy) for the breeds utilized are presented in Supplementary Data 1. Standard breed weight and lifespan were aggregated from several sources as detailed in ref. 26. We created consensus values based on the American Kennel Club and the Atlas of Dog Breeds of the World. Lifespan estimates were calculated as the average of the standard breed across sexes, compiled from numerous publications consisting primarily of surveys of multi-breed dog ages and causes of death from veterinary clinics and large-scale breed-specific surveys, which are often conducted by purebred dog associations. Sources for median-lifespan per dog breed are reported in ref. 26. We calculated the maximum lifespan for dog breeds by multiplying the median-lifespan with a factor of 1.33, i.e., $MaxLifespan = 1.33 * MedianLifespan$. Our results are qualitatively unchanged if other multipliers are used. Detailed values on the dog breeds are reported in Supplementary Data 1. Median lifespans of the 93 breeds ranged from 6.3 years (Great Dane, average adult breed weight = 64 kg) to 14.6 years (Toy Poodle, average adult breed weight = 2.3 kg). Median lifespan estimates were based on the combined findings of multiple large-scale breed health publications, utilizing the median and maximum ages for each breed.

We identified 3 dog breeds (Otterhound, $n = 4$; Weimaraner $n = 3$; Saint Bernard Dog $n = 2$) as outlier strata for which the rate of change in methylation was not meaningfully estimated according to the following exclusion criteria. In addition to their small sample sizes, the age ranges are poor all with SD (**R**) < 0.1, resulting in extreme AROCM values >0.5 (Supplementary Data 1). The remainder of the dog breeds all had AROCM values no larger than 0.34. In summary, the following criteria should be considered for our dog data or other similar data in the future.

1. Small sample size, i.e., $n < 3$.
2. Low standard deviation in relative age, i.e., SD (**R**) < 0.1.
3. Bad linear regression fit of AROCM, i.e., $R^2 < 0.2$.

### Selection of mammalian species/tissue strata

The raw data included 249 species-tissue strata from 133 unique species (Fig. 6). We selected strata with sufficient sample sizes and no influential outliers. Similar to the dog data, we excluded strata for the following reasons.

1. Small sample size of $n < 3$.
2. Low standard deviation in relative age, i.e., SD (**R**) < 0.06, to avoid strata with constant ages.
3. Strata with AROCM values out of range (estimate < −1 or >10) were omitted if the values in derived/adjacent age intervals were

not outliers. Toward this end, derived AROCMs were calculated for different age intervals within the same species/tissue stratum. For example, a severely outlying value AROCM[0, 0.1 ∗ L] was declared an outlier if both AROCM[0, 0.2 ∗ L] and AROCM[0, 0.3 ∗ L] fell within the range ( −1, 10) for the same stratum.

To obtain additional strata for the dog data, we opted for a more lenient SD(R) cutoff. Given the greater number of strata in the mammalian data compared to the dog data (229 versus 94), we selected a less strict SD(R) threshold of 0.06 instead of 0.1 to ensure sufficient strata for analysis.

For many of the 133 unique mammalian species, several tissue types were available. The species characteristics such as maximum lifespan come from an updated version of the anAge data base[7,25]. We analyzed $S$ = 229 different species/tissue strata defined on the entire age range [0, $L$] (Supplementary Data 3). Out of the 229 strata, 100 involved blood, 73 skin, 26 brain, and 15 liver. Fewer strata were available for other age ranges. For example, $S$ = 128 for the young age group (defined by [0, $0.1 ∗ L$]) and $S$ = 221 for the old age group (defined by [$0.1 ∗ L$, $L$]).

## Methylation platform

Both dog and mammalian data were generated using the same HorvathMammalMethylChip40 platform, which offers high coverage of approximately 36K conserved CpGs in mammals[28]. To minimize technical variation, all data were generated by a single lab (Horvath) using a single, consistent measurement platform. Preprocessing and normalization were performed using the SeSaMe method to define beta values for each probe[56]. The chip manifest file is available on the Gene Expression Omnibus (GEO) platform GPL28271 and on our GitHub page[28]).

## Chromatin states

Following the pan-mammalian aging study of the Mammalian Methylation Consortium, we grouped the CpGs into 54 universal chromatin states that were covered by at least 5 CpGs each[27]. These 54 chromatin states encompass those associated with both constitutive and cell-type-specific activity across a variety of human cell and tissue types[57]. In their 2022 study, Vu and Ernst employed a hidden Markov model approach to generate a universal chromatin state annotation of the human genome. This was based on data from over 100 cell and tissue types sourced from the Roadmap Epigenomics and ENCODE projects. These chromatin states are characterized in relation to 30 histone modifications, the histone variant (H2A.Z), and DNase I hypersensitivity measurements. We and others have previously found that strong age-related gain of methylation can be observed in bivalent promoter states and other states that are bound by Polycomb group repressive complex 2 (PRC2 binding sites)[27,41–43]. To facilitate a detailed analysis of PRC2 binding, we split each chromatin state into 2 subsets denoted by StateName+ and StateName- according to PRC2 binding (+ for yes and - for no). For example, the BivProm2+ is the set of 552 CpGs that reside in bivalent chromatin state 2 and are bound by PRC2 (Supplementary Data 5).

## Adjusted rate of change and adjusted correlation

The relative age $R$, defined as the ratio of age to maximum lifespan, is crucial for disentangling the relationship between rates of change and maximum lifespan (see Methods). The standard deviation of relative age, SD ($R$), reflects the sample ascertainment, collection, and design. In many real datasets, SD ($R$) varies across species due to uneven sampling, which may primarily include young or old animals in some species. This variability in SD ($R$) can dilute the signal between the rate

of change and maximum lifespan while affecting Cor(Methyl, Age), as evidenced in our simulation and empirical studies. Adjusting for SD ($R$) amplifies the inherent biological signal in both measures. Applying these formulas to the methylation data allowed us to present fundamental equations that link the rate of change in methylation in specific chromatin states (e.g. bivalent promoter regions) to maximum lifespan in mammals.

Here we present a mathematical formalism that links three measurements: i) the rate of change in the biomarker across the life course, ii) the Pearson correlation between age and the biomarker, and iii) the standard deviation of relative age. In most empirical datasets, the standard deviation of relative age is correlated with the Pearson correlation between age and the biomarker (Supplementary Fig. 2), which reflects the idiosyncrasies of the sample collection. The standard deviation of age has a confounding effect on both the rate of change and the correlation between a biomarker and age. To study and eliminate this confounding effect, we introduce two concepts: adjusted rate of change and adjusted Pearson correlation. We present mathematical propositions describing the conditions under which strong relationships between the rate of change and lifespan can be observed.

In the following, we derive general equations that link the rate of change (also known as gradient or slope) of any continuous biomarker of aging (denoted as $M \in R$) to the species maximum lifespan. For example, $M$ could denote mean methylation in a particular chromatin state. Assume $\mathbf{M} = (M_1, \ldots M_n)$ and $\mathbf{A} = (A_1, \ldots, A_n)$ are two numeric vectors of $n$ samples for the biomarker $M$ and the Age variable. We will be using the following definitions surrounding the sample mean, sample variance and standard deviation, coefficient of variation, sample covariance, and Pearson correlation.

$$
\begin{aligned}
\overline{\mathbf{M}} &\doteq \frac{1}{n}\sum_{i=1}^{n} M_i, \\
\text{Var}(\mathbf{M}) &\doteq \frac{1}{n}\sum_{i=1}^{n}(M_i - \overline{\mathbf{M}})^2, \\
\text{SD}(\mathbf{M}) &\doteq \sqrt{\text{Var}(\mathbf{M})} \\
\text{CoefVar}(\mathbf{M}) &\doteq \frac{\text{SD}(\mathbf{M})}{\overline{\mathbf{M}}}, \\
\text{Cov}(\mathbf{M},\mathbf{A}) &\doteq \frac{1}{n}\sum_{i=1}^{n}(M_i - \overline{\mathbf{M}})(A_i - \overline{\mathbf{A}}), \\
\text{Cor}(\mathbf{M},\mathbf{A}) &\doteq \frac{\text{Cov}(\mathbf{M},\mathbf{A})}{\sqrt{\text{Var}(\mathbf{M}) * \text{Var}(\mathbf{A})}}.
\end{aligned} \tag{10}
$$

Next, we define the rate of change, ROC($\mathbf{M}$; $\mathbf{A}$), as the change in $M$ resulting from a 1-year increase in age (calendar age in units of years). Statistically speaking, the rate of change, ROC($\mathbf{M}$; $\mathbf{A}$), is the slope/coefficient $\beta_1$ in the univariate linear regression model below,

$$
M_i = \beta_0 + \beta_1 A_i + \epsilon_i,
$$

where the index $i$ refers to the $i$-th tissue sample and the expected value of the error term $\epsilon_i$ is assumed to be zero. The rate of change can be estimated by the least squares or the maximum likelihood estimator, $\widehat{\beta}_1$. Furthermore, it can be expressed in terms of the Pearson correlation coefficient and standard deviations as follows

$$
\text{ROC}(\mathbf{M}; \mathbf{A}) \doteq \widehat{\beta}_1 = \frac{\text{Cor}(\mathbf{M},\mathbf{A})\text{SD}(\mathbf{M})}{\text{SD}(\mathbf{A})}. \tag{11}
$$

To arrive at a unit-less biomarker, which lends itself to comparisons with other biomarkers, we standardize $\mathbf{M}$ to have mean zero and

standard deviation one, by scaling it as below,

$$ScaledM_i \doteq \frac{M_i - \overline{\mathbf{M}}}{\mathrm{SD}(\mathbf{M})}.$$

In our dataset, we do not observe a significant correlation between SD($\mathbf{M}$) and lifespan ($L$), see Supplementary Fig. 16. Using SD($\mathbf{ScaledM}$) = 1, equation (11) becomes

$$\mathrm{ROC}(\mathbf{ScaledM}; \mathbf{A}) \doteq \widehat{\beta}_1 = \frac{\mathrm{Cor}(\mathbf{ScaledM}, \mathbf{A})}{\mathrm{SD}(\mathbf{A})} = \frac{\mathrm{Cor}(\mathbf{M}, \mathbf{A})}{\mathrm{SD}(\mathbf{A})} \quad (12)$$

where the latter equation used the fact that the Pearson correlation, Cor, is invariant with respect to linear transformations. To reveal the dependence on species maximum lifespan, it is expedient to define relative age as the ratio of age and maximum lifespan:

$$R_i = \frac{A_i}{L}. \quad (13)$$

Since the standard deviation is the square root of the variance, one can easily show that SD($\mathbf{A}$) = SD($\mathbf{R}$) $* L$. Combining equations (12) and (13) results in

$$\mathrm{ROC}(\mathbf{ScaledM}; \mathbf{A}) = \frac{\mathrm{Cor}(\mathbf{M}, \mathbf{A})/\mathrm{SD}(\mathbf{R})}{L}. \quad (14)$$

Since Pearson's correlation is scale-invariant, the following equality holds and we will use them interchangeably, Cor($\mathbf{M}$, $\mathbf{A}$) = Cor($\mathbf{ScaledM}$, $\mathbf{A}$) = Cor($\mathbf{M}$, $\mathbf{R}$).

**Proposition 1.** Relationship between ROC and Lifespan If the following condition holds across all strata,

$$\mathrm{Cor}(\mathbf{M}, \mathbf{A})/\mathrm{SD}(\mathbf{R}) \approx constant, \quad (15)$$

then equation (14) implies

$$\mathrm{ROC}(\mathbf{ScaledM}; \mathbf{A}) \approx \frac{constant}{L}. \quad (16)$$

Due to sampling bias and uneven distributions of relative age, the strong condition (15) is usually not satisfied (see, for example, Supplementary Fig. 2f). We propose a simple adjustment to formulate a weaker, more realistic assumption that leads to a conclusion similar to equation (16). To this end, we rewrite equation (14) as follows:

$$\mathrm{ROC}(\mathbf{ScaledM}; \mathbf{A}) \times \mathrm{SD}(\mathbf{R})^{1-p} = \frac{\mathrm{Cor}(\mathbf{M}, \mathbf{A})/\mathrm{SD}(\mathbf{R})^p}{L}, \quad (17)$$

which multiplies both sides by SD($\mathbf{R}$)$^{1-p}$ with a power parameter $p$. Next we define:

$$Adj.\mathrm{ROC}(\mathbf{ScaledM}; \mathbf{A}, p) \doteq \mathrm{ROC}(\mathbf{ScaledM}|\mathbf{A}) \times \mathrm{SD}(\mathbf{R})^{1-p},$$
$$\mathrm{Adj.Cor}(\mathbf{M}|\mathbf{R}, p) \doteq \frac{\mathrm{Cor}(\mathbf{R}, \mathbf{M})}{\mathrm{SD}(\mathbf{R})^p}. \quad (18)$$

Note that if SD($\mathbf{R}$) remains constant across strata, indicative of a perfect design, the adjustment essentially involves multiplying or dividing by a constant, irrespective of the power $p$. This means the adjustment leaves the relationship between ROC and lifespan unchanged. On the other hand, if SD($\mathbf{R}$) fluctuates across strata-indicative of an imperfect study the adjustments have the potential to enhance the signal. Further, note that Adj.ROC becomes the standard definition of the ROC with $p$ = 1. On the other hand, $p$ = 0 implies that Adj.Cor($\mathbf{M}|\mathbf{R}, p$) =

Cor($\mathbf{M}$, $\mathbf{R}$). We introduce this terminology for several reasons. To begin with, equation (17) can be succinctly written as follows

$$Adj.\mathrm{ROC}(\mathbf{ScaledM}; \mathbf{A}, p) = \frac{\mathrm{Adj.Cor}(\mathbf{M}|\mathbf{R}, p)}{L}. \quad (19)$$

The following material outlines the specific conditions required for the validity of the equation below:

$$\mathrm{Adj.ROC} \approx \frac{c}{L},$$

where $c$ is a constant. Here, the approximation sign $\approx$ indicates a strong linear correlation across strata when assessed on a logarithmic scale. We start with the log-transformed version of equation (19):

$$\log(Adj.\mathrm{ROC}(\mathbf{ScaledM}; \mathbf{A}, p)) = \log(\mathrm{Adj} \cdot \mathrm{Cor}(\mathbf{M}|\mathbf{R}, p)) - \log(L), \quad (20)$$

where we assume that the natural logarithm (log) is applicable, i.e., the adjusted ROC and the adjusted correlation take on positive values.

The above-mentioned definitions and equations apply to each stratum (e.g., each dog breed). Assuming there are $S$ total strata, we introduce a superscript in various quantities, e.g., we write $L^{(s)}$, and Adj.Cor($\mathbf{M}^{(s)}|\mathbf{R}^{(s)}, p$), where $s$ = 1, 2, . . . , $S$. Define the following 3 vectors that have $S$ components each

$$\mathbf{\log.L} = \left(\log(L^{(1)}), \log(L^{(2)}, ..., \log(L^{(S)})\right)$$
$$\mathbf{\log.Adj.Cor}(p) = \left(\log\left(\mathrm{Adj.Cor}(\mathbf{M}^{(s)}|\mathbf{R}^{(s)}, p)\right)\right)_{1 \le s \le S}$$
$$\mathbf{\log.Adj.ROC}(p) = \left(\log\left(Adj.\mathrm{ROC}(\mathbf{ScaledM}^{(s)}; \mathbf{A}^{(s)}, p)\right)\right)_{1 \le s \le S}.$$

For each vector on the left-hand side, we can form the sample mean and sample variances across $S$ strata,

$$\overline{\mathbf{\log.Adj.Cor}} = \frac{1}{S} \sum_{s=1}^{S} \log(\mathrm{Adj.Cor}(\mathbf{M}^{(s)}|\mathbf{R}^{(s)}))$$
$$\mathrm{Var}(\mathbf{\log.Adj.Cor}) = \frac{1}{S} \sum_{s=1}^{S} \left(\log(\mathrm{Adj.Cor}(\mathbf{M}^{(s)}|\mathbf{R}^{(s)})) - \overline{\mathbf{\log.Adj.Cor}}\right)^2$$

We will present several propositions and outline their proofs. In some cases, we provide only a rough outline, as exact derivations would require more complex formalism. The following critical condition states that lifespan does not correlate with adjusted age correlation on the log scale:

(**C1**)

$$\mathrm{Cor}(\mathbf{\log.L}, \mathbf{\log.Adj.Cor}(p)) = 0. \quad (21)$$

Condition Cor($\mathbf{\log.L}$, $\mathbf{\log.Adj.Cor}(p)$) = 0 holds when species lifespan and the Adj.Cor($p$) are independent across strata. Our methylation data suggest that this condition is approximately satisfied for certain chromatin states (Fig. 7).

**Proposition 2.** If (C1) holds, then

$$\mathrm{Cor}(\mathbf{\log.L}, \mathbf{\log.Adj.ROC(p)}) = \frac{-1}{\sqrt{1 + \mathrm{Var}(\mathbf{\log.Adj.Cor})/\mathrm{Var}(\mathbf{\log.L})}}$$

**Proof.** Denote vectors $\mathbf{x}$ = $\mathbf{\log.L}$ and $\mathbf{y}$ = $\mathbf{\log.Adj.ROC}$. With equation (20) we find that the covariance

$$\mathrm{Cov}(\mathbf{x}, \mathbf{y}) = \mathrm{Cov}(\mathbf{\log.L}, \mathbf{\log.Adj.Cor}) - \mathrm{Cov}(\mathbf{\log.L}, \mathbf{\log.L})$$

By assumption, the first term is zero, which entails that

$$\text{Cov}(\mathbf{x}, \mathbf{y}) = -\text{Cov}(\log.\mathbf{L}, \log.\mathbf{L}) = -\text{Var}(\mathbf{x})$$

Similarly, $\text{Cov}(\log.\mathbf{L}, \log.\mathbf{Adj.Cor}) = 0$ implies that

$$\text{Var}(\mathbf{y}) = \text{Var}(\log.\mathbf{L}) - 2*\text{Cov}(\log.\mathbf{L}, \log.\mathbf{Adj.Cor}) + \text{Var}(\log.\mathbf{Adj.Cor})$$
$$= \text{Var}(\log.\mathbf{L}) + \text{Var}(\log.\mathbf{Adj.Cor})$$

Thus, the assumption implies that

$$\text{Cor}(\mathbf{x}, \mathbf{y}) = \frac{\text{Cov}(x, y)}{\sqrt{\text{Var}(\mathbf{x})\text{Var}(\mathbf{y})}}$$
$$= -\frac{\text{Var}(\mathbf{x})}{\sqrt{\text{Var}(\mathbf{x})(\text{Var}(\mathbf{x}) + \text{Var}(\log.\mathbf{Adj.Cor}))}}$$
$$= -\frac{1}{\sqrt{1 + \text{Var}(\log.\mathbf{Adj.Cor})/\text{Var}(\log.\mathbf{L})}}.$$

The following proposition is a direct consequence of Proposition 2.

**Proposition 3.** If (C1) holds and the ratio

$$\text{Ratio}(p) = \frac{\text{Var}(\log.\mathbf{Adj.Cor}(p))}{\text{Var}(\log.\mathbf{L})} \approx 0, \quad (22)$$

then

$$\log.\mathbf{Adj.ROC} \approx \overline{\log.\mathbf{Adj.Cor}} - \log.\mathbf{L}.$$

Proposition 3 implies that Lifespan and Adj.ROC follows a nearly perfect inverse linear correlation on the log scale ($\text{Cor} \approx -1$) if $\text{Var}(\log.\mathbf{Adj.Cor}) \ll \text{Var}(\log.\mathbf{L})$. The latter condition is typically satisfied in real data as the range of lifespans across strata is often much larger than the Adj.Cor values, which is the case for our data from the mammalian methylation consortium.

**Proof.** Proposition 2, combined with the assumption that $\text{Ratio}(p) \approx 0$, leads to the conclusion that $\text{Cor}(\log.\mathbf{Adj.ROC}, \log.\mathbf{L}) \approx -1$. Given that a Pearson correlation nearing negative one indicates an almost perfect linear relationship, this finalizes the proof.

We are now ready to state the main proposition.

**Proposition 4.** The linear relationship between log.Adj.ROC and log.L If (C1, equation (21)) holds and the squared coefficient of variation in Adj.Cor($p$) is much smaller than the squared coefficient of variation in $L$, i.e.,

$$\text{Ratio}(p) = \frac{\text{CoefVar}(\text{Adj.Cor}(p))^2}{\text{CoefVar}(L)^2} \approx 0 \quad (23)$$

then

$$\log.\mathbf{Adj.ROC} \approx \overline{\log.\mathbf{Adj.Cor}} - \log.\mathbf{L}. \quad (24)$$

**Proof.** In the following, we will show that the assumption (equation (23)) implies equation (22) in Proposition 3. We will use the following Delta method approximation for computing the variance of $f(X)$ of a random variable $X$,

$$\text{Var}(f(X)) \approx f'(\text{E}(X))^2 \text{Var}(X),$$

where $\text{Var}(X)$ and $\text{E}(X)$ denote the variance and expectation of $X$, respectively. With $f(x) = \log(x)$, $f'(x) = 1/x$ and $X = \text{Adj.Cor}(p)$, the above approximation results in

$$\text{Var}(\log(\text{Adj.Cor}(p))) \approx \frac{\text{Var}(\text{Adj.Cor}(p))}{\text{E}(\text{Adj.Cor}(p))^2} = \text{CoefVar}^2(\text{Adj.Cor}(p))$$

where $\text{CoefVar}(\cdot)$ denotes the coefficient of variation. Analogously, we have

$$\text{Var}(\log(L)) \approx \frac{\text{Var}(L)}{\text{E}(L)^2} = \text{CoefVar}^2(L).$$

Therefore, (23) implies (22) and concludes the proof.

Condition (23) is approximately satisfied in the mammalian data and the dog data: in the mammalian data, $\text{CoefVar}(\text{Adj.Cor}(p)) = 0.28$ and $\text{CoefVar}(L) = 0.91$ resulting in $Ratio(p) = 0.095$. In the dog breed data, $\text{CoefVar}(\text{Adj.Cor}(p)) = 0.12$ and $\text{CoefVar}(L) = 0.16$ resulting in $Ratio(p) = 0.56$. The judicious choice of the adjustment power $p$ resulted in lower coefficients of variation, as can be seen in the comparison with the unadjusted values: $\text{CoefVar}(\text{Cor/SD}) = 0.68$ for the mammalian data and 0.24 for the dog data.

Exponentiating both sides of equation (24), we arrive at

$$\text{Adj.ROC}^{(s)} \approx \frac{c(p)}{L^{(s)}} \quad (25)$$

where $c(p) = \exp\left(\overline{\log.\mathbf{Adj.Cor}}\right)$ is some constant. The choice of the parameter $p$ will be discussed in the following.

### Criteria for choosing the power $p$ in the adjustment

Our aforementioned equations utilize the parameter $p$, which underlies our definitions of the adjusted correlation and the adjusted ROC. Choosing $p = 1$ results in standard (non-adjusted) versions of the ROC, but opting for a lower value of $p$ can be advantageous for the following three reasons: First, Proposition 4 states that a strong linear relationship between **log.Adj.ROC** and l*og. L* holds if $p$ is chosen to minimize the coefficient of variation function: $C(p) = \text{CoefVar}(\text{Adj} \cdot \text{Cor}(p))$. Since the coefficient of variation is sensitive to outliers, we find it expedient to use a robust alternative known as the quartile coefficient of dispersion (QCOD):

$$\text{QCOD}(p) \doteq \frac{Q_3(p) - Q_1(p)}{Q_3(p) + Q_1(p)}. \quad (26)$$

where $Q_1(p)$ and $Q_3(p)$ denote the first and third quartile of the distribution of Adj.Cor($p$). In our empirical studies, we chose $p$ so that it minimized QCOD (equation (26)), i.e.,

$$p_{optimal} = \arg\min_p \text{QCOD}\left[\log(\text{Adj.Cor}(p))\right] \quad (27)$$

Using the QCOD-based criterion, we determined $p_{optimal} = 0.1$ for our dog data and $p_{optimal} = 0.25$ for our mammalian methylation dataset (refer to Supplementary Fig. 1). Had we employed the coefficient of variation in place of the QCOD, our choice of $p$ would have been consistent across both datasets, as depicted in Supplementary Fig. 1. This alignment between the coefficient of variation and QCOD is well-documented in statistical literature, as cited in[58,59]. The second reason for choosing the power $p$ relates to an undesirable correlation between the age correlation Cor(**M**, **A**) and the standard deviation of relative age, SD(**R**) (Supplementary Fig. 2). Our simulation studies suggest that this positive correlation results from an imperfect sample ascertainment/study design. This can be mitigated against by choosing $p$ so that the adjusted age correlation Adj.Cor($M|\mathbf{R}$, $p$) exhibits a weaker correlation with SD(**R**). In the

mammalian data, $p = 0.25$ leads to a non-significant correlation between Adj.Cor($M|\mathbf{R}$, $p$) and SD($\mathbf{R}$) (Supplementary Fig. 2g, h). Third, our simulation studies, designed to emulate our mammalian lifespan data, indicate that with large sample sizes per stratum, Adj.Cor($M|\mathbf{R}$, $p$) converges to a value close to 1.0 for $p = 0.25$ (Supplementary Note 2). This value of 1.0 significantly simplifies the equations. We use simulations to study the relationship between the age correlation and the standard deviation of relative age as a function of the data ascertainment (Supplementary Fig. 17). Further, we explore the effect of the adjustment power $p$ in Supplementary Fig. 18. The coefficient of variation displays a U-shape when the power increases, hence a minimum is achievable. The optimal adjustment power is achieved at 0.25 for most cases. Overall, these results suggest that $p = 0.25$ is a good choice for our mammalian methylation study.

### Relation between AROCM$_{young}$ and AROCM$_{old}$

Here, we provide an outline on how to derive a relationship between the rate of change in young animals and that in older ones in the s-th species-tissue stratum, i.e.,

$$\text{AROCM}_{young}^{(s)} = c * \text{AROCM}_{old}^{(s)}. \quad (28)$$

where $c$ denotes a constant. We start out by commenting on our definition of relative age. When dealing with prenatal samples (whose chronological ages take negative values), it can be advantageous to slightly modify the definition of relative age as $R = \frac{A+GT}{L+GT}$, by including gestation time (GT) to avoid negative relative ages. For simplicity, we will assume that our data only contains postnatal samples, allowing us to define relative age as $R = \frac{A}{L}$. Empirically, we find that the non-linear relationship between ScaledM and relative age in each stratum can be approximated using the following function:

$$\begin{aligned} \text{ScaledM}_i^{(s)} &= f(R; \gamma) \\ &= \gamma_0^{(s)} + \gamma_1^{(s)} g(R), \end{aligned} \quad (29)$$

where $\gamma^{(s)} = (\gamma_0^{(s)}, \gamma_1^{(s)})$ are stratum-specific constants. Our empirical studies demonstrate that the following log-linear function fits the data quite well.

$$g(R) = \begin{cases} 10R - 1 & R \geq 0.1 \\ \log(10R) & R < 0.1 \end{cases} \quad (30)$$

Note that the first derivative of $g()$ is given by

$$g'(R) = \begin{cases} 10 & R \geq 0.1 \\ 1/R & R < 0.1 \end{cases} \quad (31)$$

Assuming a linear relationship between ScaledM and $A$ (equation (3)) and a suitably chosen midpoint $A_0$, one can approximate AROCM as follows

$$\begin{aligned} \text{AROCM} &= \frac{\Delta \text{ScaledM}}{\Delta A} \\ &\approx \frac{d}{dA}(\text{ScaledM}^{(s)})|_{A_0} \\ &= \frac{d}{dA} f(\frac{A}{L})|_{A_0} \\ &= \frac{d}{dR} f(R)|_{R_0} \frac{1}{L} \\ &= \gamma_1 * g'(R_0) \frac{1}{L} \end{aligned} \quad (32)$$

where $R_0 = A_0/L$ represents the relative age of a young or old individual, and we used the chain rule of calculus. We define the AROCM in

young and old animals as the first derivative evaluated at $A_{young}$ and $A_{old}$, respectively. These ages should be chosen so that the corresponding relative ages $R_{young}$ and $R_{old}$ take on values $<0.1$ and $>0.1$, respectively. With equations (31) and (32), we find

$$\begin{aligned} \text{AROCM}_{young} &= \gamma_1 \times \frac{1}{R_{young}L} \\ \text{AROCM}_{old} &= \gamma_1 \times \frac{10}{L} \end{aligned} \quad (33)$$

With superscripts denoting the s-th species-tissue stratum, it implies the following linear relationship between the two aging rates

$$\text{AROCM}_{young}^{(s)} = \frac{1}{10R_{young}^{(s)}} \times \text{AROCM}_{old}^{(s)}. \quad (34)$$

Since the young groups for all strata are defined with the same cutoff of $R = 0.1$, $R_{young}^{(s)}$ would take similar values across all strata, which implies that $\text{AROCM}_{young}^{(s)} = c \times \text{AROCM}_{old}^{(s)}$. Empirically, we can verify the latter relationship (Fig. 4). Across species-tissue strata, we find that $c$ has a mean value of 7.33 and a standard deviation of 6.8.

There is an analogous relationship between AROCM$_{young}$ and AROCM$_{old}$ when a different function $g$ is used. For instance, when the function $g_2(R) = \log(10R)$ (for all values of $R$) is used, we can derive the relationship

$$\text{AROCM}_{young}^{(s)} = \frac{R_{old}^{(s)}}{R_{young}^{(s)}} \times \text{AROCM}_{old}^{(s)}$$

Consequently, $\text{AROCM}_{young}^{(s)}$ is still proportional to $\text{AROCM}_{old}^{(s)}$, provided that the ratio $\frac{R_{old}^{(s)}}{R_{young}^{(s)}}$ remains approximately constant across all strata. We compared $g(R)$ and $g_2(R)$ in our mammalian data as shown in Supplementary Fig. 19. The median correlation across all species is the highest using $g(R)$ ($r = 0.76$), compared to the original relative age ($r = 0.73$) and $g_2(R)$ ($r = 0.74$).

### Reporting summary
Further information on research design is available in the Nature Portfolio Reporting Summary linked to this article.

## Data availability
The individual-level data from the Mammalian Methylation Consortium can be accessed from several online locations. All data from the Mammalian Methylation Consortium are posted on Gene Expression Omnibus (complete dataset, GSE223748[25,27]). Additional details can be found in Supplementary Note 3. The mammalian methylation array is available through the non-profit Epigenetic Clock Development Foundation (https://clockfoundation.org/data-tools/, data available at https://www.ncbi.nlm.nih.gov/geo/query/acc.cgi?acc= GPL28271). Numerical results generated in this study are provided in the Supplementary Information/Source Data file. Source data are provided in this paper.

## Code availability
The mammalian methylation array manifest files, genome annotations of CpG sites including chromatin state information can be found on Zenodo (https://doi.org/10.5281/zenodo.7574747) and Github https://github.com/shorvath/MammalianMethylationConsortium/tree/v2.0.0. The R scripts and software are being distributed through Zenodo/GitHub at github.com/feizhe/FundamentalEquations(10.5281/zenodo.12610965).

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

## Acknowledgements
This work was supported by the Paul G. Allen Frontiers Group (S.H.) and a grant from Open Philanthropy (S.H.).

## Author contributions
S.H. and Z.F. conceived the study and developed the mathematical framework. Z.F., J.Z., A.H., A.T.L., S.H. conducted the the statistical analyses. All authors contributed to editing the article and interpreting the data.

## Competing interests
The Regents of the University of California filed a patent application (publication number WO2020150705) related to the mammalian methylation array platform for which S.H. is a named inventor. S.H. is a founder of the non-profit Epigenetic Clock Development Foundation, which has licensed several patents from his employer UC Regents, and distributes the mammalian methylation array. S.H. works for Altos Labs, Inc. The other authors declare no conflicts of interest.
