## [Peer Review File · Nature Communications]

Fundamental equations linking methylation dynamics to maximum lifespan in mammalsEditorial Note: This manuscript has been previously reviewed at another journal that is not operating a transparent peer review scheme. This document only contains reviewer comments and rebuttal letters for versions considered at Nature Communications.

Reviewers' Comments:

Reviewer #1:

Remarks to the Author:

First, to address a concern explicitly raised by the authors: I am not a co-author on the Crofts et al. paper. In fact, I do not work on DNA methylation clocks at all. Thus, I am definitely not a competitor. Second, the authors have dramatically improved the presentation of their results. The text reads much better and the key novelties of the work are presented much more clearly.

Major points

(1)

Can the authors reproduce their main findings by reducing the data to strata with similar SD(R)? Note that SD(R) of some species can possibly be made more similar by downsampling the data in a suitable way.

(2)

I have issues with the sections "Development versus aging" (in the Discussion) and "Relation between AROCM_{young} and AROCM_{old}" (in the Methods).

"Importantly, we demonstrate that the the rate of change in young is proportional to that in old animals (Figure 4). This latter discovery can be understood from two perspectives: mathematical and biological." (Note: there is one 'the' too many.)

I think this statement is misleading. Biology tells us that the two measures should be correlated within the same species. Biologically this is plausible, simply because the same mechanisms controlling epigenetic states are at work in young and old individuals of the same species. (The cells have the same genes after all.) However, I do not understand why this would be a mathematical necessity. Instead, the section "Relation between AROCM_{young} and AROCM_{old}" motivates why AROCM_{young} and AROCM_{old} should be linearly related under the assumption(!) that the molecular process itself does not change. The fact that AROCM_{young} and AROCM_{old} are related by a constant relies on the assumption(!) that γ_1 is constant (i.e. the same) in young and old (equation 31).

If one assumes that the rates are the same in young and old, then of course the two measures will be correlated. One can turn this around: under which circumstances would the two measures be uncorrelated? Only if the performance/fidelity of quality control processes maintaining epigenetic states in young and old individuals were uncorrelated. That however, is a pure biological interpretation and not a mathematical.

Thus, it should be made explicit in the Discussion that biology suggests that the process itself does not dramatically change within a species and that mathematics then explains (under that assumption!) why AROCM_{young} and AROCM_{old} should be linked by a constant (as opposed to having some other non-linear, but monotonic relationship).

In addition, in equation 28 I don't understand why different approximations are being used for old and young. The fact that a log-linear model works well suggests that some kind of exponential process is at work. Thus, why can $g(R)$ not be approximated by $\log(10R)$ in both, young and old?

However, this is probably just a minor point, not affecting the conclusions.

Minor points

(3) Abstract:

"we propose a robust approach that identifies the strong inverse relationship for certain biomarkers"

Consider the following alternative:

"we propose a robust approach that identifies a strong inverse relationship of lifespan with certain biomarkers"

(4) Page 4:

"our mathematical formalist predicts an inverse relationship between Adj.AROCM and Lifespan"
Repeated statement.

(5) Figure S14:

Panels a and b are not explicitly referenced in the caption. Second, what are the vertical red bars in panels a and b showing?

(6) Discussion:

"These findings collectively underscore the significance of selecting an appropriate chromatin context for defining the AROCM."

This wording would be appropriate, if the goal was to predict lifespan. In that case one would want to choose CpGs that are most strongly correlated with lifespan. However, the spin of this publication is more mechanistic: it directly addresses the process of DNA methylation. Thus, analyzing e.g. TxEx4 is not 'inappropriate'. Instead, that analysis reveals that those CpGs seem to be under different regulatory influence. Hence, I would prefer the following wording instead:

"These findings collectively underscore that the chromatin context substantially affects the relationship between AROCM and 1/Lifespan."

Reviewer #2:

Remarks to the Author:

In response to the reviewer's comments, the authors have removed their unsubstantiated claims and corrected the numerous errors in the formulas. The major claim that methylation negatively scales with maximum lifespan has been already published by Crofts et al. in Nature Aging 2024. As outlined in the original reviewer comments, this paper was preprinted already earlier and is still ignored by the authors with the argument that Dr. Horvath described the idea in a TEDx video. It is unscholarly to refuse credence to authors who published the scientific paper describing the finding. Taken together, the advance provided by the current manuscript in comparison to the state of the art is unclear. The listing of the mathematical formulas could be useful to bioinformatics specialists in the methylation clock field.

Additional specific comments:

- The article is still poorly written. Some key points are written in a convoluted way or point to the middle of the methods section without more explanation. While the main text doesn't need to contain all the mathematical details it should guide the reader more clearly and make the advance understandable without having to read the whole methods section.
- One of the main aim is to show that it is beneficial to correct for imperfect sample biases. This is interesting to note and indeed a lot of people in the field might not directly be aware of this. How would the (un-)adjusted AROCM vs. log lifespan look like if only considering strata with roughly equal SDs? If the adjustment is working without introducing artificial biases the results should not change. Reviewer 1 does have a valid point on the circular reasoning, so the statement of overfitting in the discussion is indeed important.

Detailed Response to Reviewer #1:

We will highlight the reviewer comments in bold.

First, to address a concern explicitly raised by the authors: I am not a co-author on the Crofts et al. paper. In fact, I do not work on DNA methylation clocks at all. Thus, I am definitely not a competitor.

Second, the authors have dramatically improved the presentation of their results. The text reads much better and the key novelties of the work are presented much more clearly.

Response by the authors: Thank you immensely for your encouraging and thoughtful review. We are especially appreciative of your meticulous examination of our material and your insightful feedback, which has been instrumental in enhancing the clarity and effectiveness of our presentation. Your contributions have significantly improved our work.

Major points

(1) Can the authors reproduce their main findings by reducing the data to strata with similar SD(R)? Note that SD(R) of some species can possibly be made more similar by downsampling the data in a suitable way.

Response: Yes, we have fully implemented this suggestion.

In response to the recommendation, Supplementary figure S5 has been added to closely replicate the insights of main Figure 5, elucidating the relationship between AROCM and lifespan across various strata characterized by similar SD(R) values. We have categorized these strata by the range of SD(R) values, specifically into three distinct strata: (0, 0.2), (0.2, 0.4), and (0.4, 0.6), to facilitate a stratified analysis.

The following was added to subsection "Adjusted AROCM approximates the inverse of maximum lifespan" in the Results:

"Our findings underscore the efficacy of the adjusted AROCM in addressing the issue of strata with widely varying values of $\text{sd}(R)$. Traditionally, one might handle the variability in $\text{sd}(R)$ by limiting the analysis to strata that have approximately similar $\text{sd}(R)$ values. To demonstrate that our principal findings are consistent using this conventional method, we conducted the same analysis on strata with comparable $\text{sd}(R)$ values. The results were compelling: we observed strong correlations between AROCM and $1/\text{Lifespan}$, with correlation coefficients of $r = 0.99$ for strata within $\text{sd}(R) \in (0.2, 0.4)$, and $r = 0.97$ for strata within $\text{sd}(R) \in (0.4, 0.6)$, respectively (see Supplementary Figure S5). Furthermore, a stratified analysis reaffirmed our key findings from the dog breed study (refer to Supplementary Figure S6)."

(2) I have issues with the sections “Development versus aging” (in the Discussion) and “Relation between AROCM_{young} and AROCM_{old}” (in the Methods). “Importantly, we demonstrate that the the rate of change in young is proportional to that in old animals (Figure 4). This latter discovery can be understood from two perspectives: mathematical and biological.” (Note: there is one ‘the’ too many.)

Response: We have corrected the typo of duplicated “the”.

I think this statement is misleading. Biology tells us that the two measures should be correlated within the same species. Biologically this is plausible, simply because the same mechanisms controlling epigenetic states are at work in young and old individuals of the same species. (The cells have the same genes after all.) However, I do not understand why this would be a mathematical necessity.

Instead, the section “Relation between AROCM_{young} and AROCM_{old}” motivates why AROCM_{young} and AROCM_{old} should be linearly related under the assumption(!) that the molecular process itself does not change. The fact that AROCM_{young} and AROCM_{old} are related by a constant relies on the assumption(!) that γ_1 is constant (i.e. the same) in young and old (equation 31).

If one assumes that the rates are the same in young and old, then of course the two measures will be correlated. One can turn this around: under which circumstances would the two measures be uncorrelated? Only if the performance/fidelity of quality control processes maintaining epigenetic states in young and old individuals were uncorrelated. That however, is a pure biological interpretation and not a mathematical.

Thus, it should be made explicit in the Discussion that biology suggests that the process itself does not dramatically change within a species and that mathematics then explains (under that assumption!) why AROCM_{young} and AROCM_{old} should be linked by a constant (as opposed to having some other non-linear, but monotonic relationship).

Response:

Thank you for providing these insightful comments. We are in full agreement. Accordingly, we have revised the paragraph to better capture and reflect your points.

Previous research has shown that rates of developmental change in methylation exceed those observed post-puberty in humans \citep{horvath2013dna, farrell2020epigenetic}. Our extensive mammalian dataset supports this observation across a broad range of species, as shown in Figure \ref{fig.AROCM.Young.versus.old}. Notably, we have established a proportional relationship between the rates of change in young and old animals, a finding consistent across species (Figure \ref{fig.AROCM.Young.versus.old}). The underlying principle of this proportionality is mathematically articulated in equation \ref{eq:AROCMyoungvsold}, which can be derived from the biological premise that a uniform epigenetic maintenance mechanism regulates methylation levels across the lifespan of a species. More specifically, this proportionality emerges from the hypothesis of a continuous, although potentially non-linear, increase in cytosine methylation at specific genomic locations (bivalent promoters) from development through old age. This has been explicitly formulated in our life course equations \ref{lm_arocm_logli} and

$\text{ref}\{eq:loglinearrelativeage\}$, assuming a non-linear relationship between scaled methylation (ScaledM) and relative age.

From this standpoint, if we consider that identical epigenetic maintenance processes govern methylation increases in both young and old animals, it logically follows that $\text{ARO}\text{CM}_{\text{young}}$ and $\text{ARO}\text{CM}_{\text{old}}$ would be correlated across different species. Conversely, a lack of correlation would imply fundamentally distinct epigenetic regulation mechanisms at different life stages.

Biologically, the proportionality between $\text{ARO}\text{CM}_{\text{young}}$ and $\text{ARO}\text{CM}_{\text{old}}$ lends support to deterministic aging theories, which propose a connection between developmental changes and those that occur later in life $\text{citep}\{de2005genomes,blagosklonny2013aging,horvath2018dna,de2023ageing,gems2022hyperfunction\}$. Our theoretical model explains, under the premise that young and old animals share the same epigenetic maintenance mechanisms, why $\text{ARO}\text{CM}_{\text{young}}$ and $\text{ARO}\text{CM}_{\text{old}}$ are linked by a multiplicative constant, rather than a different type of non-linear but monotonic relationship.

In addition, in equation 28 I don't understand why different approximations are being used for old and young. The fact that a log-linear model works well suggests that some kind of exponential process is at work. Thus, why can $g(R)$ not be approximated by $\log(10R)$ in both, young and old?

However, this is probably just a minor point, not affecting the conclusions.

Response: This is a good point. We added a supplement figure to show that the log linear approximation is slightly better than the log transformation. See below for more details. As the reviewer suggests, assuming a different form of the function, denoted as $g_2(R)=\log(10R)$ in both and young, leads to a similar conclusion. In response we have inserted the following sentences.

“There is an analogous relationship between $\text{ARO}\text{CM}_{\text{young}}$ and $\text{ARO}\text{CM}_{\text{old}}$ when a different function g is used. For instance, when the function $g_2(R) = \log(10R)$ (for all values of R) is used, we can derive the relationship

$$\text{ARO}\text{CM}^{(s)}_{\text{young}} = \frac{R^{(s)}_{\text{old}}}{R^{(s)}_{\text{young}}} \times \text{ARO}\text{CM}^{(s)}_{\text{old}}$$

Consequently, $\text{ARO}\text{CM}^{(s)}_{\text{young}}$ is still proportional to $\text{ARO}\text{CM}^{(s)}_{\text{old}}$, provided that the ratio $\frac{R^{(s)}_{\text{old}}}{R^{(s)}_{\text{young}}}$ remains approximately constant across all strata.

In our study, we prioritized examining $g(R)$ over $g_2(R)$ because $g(R)$ demonstrated a more accurate fit in our mammalian dataset. A key metric supporting this choice is the median Pearson correlation coefficient between ScaledM and the inverse of $g(R)$, denoted as $g^{-1}(R)$, yielded a median Pearson correlation of $r=0.76$. This correlation slightly surpasses the correlation observed between ScaledM and the inverse of $g_2(R)$,

$g_{-1}(R)$, which stands at $r=0.74$. Furthermore, both correlations were slightly higher than the correlation between ScaledM and (untransformed) R ($r=0.73$), as illustrated in Supplementary Figure S14. These findings show that $g(R)$ offers a superior fit for the mammalian methylation data. However, we urge the research community to explore and identify transformations that might yield even better fits.

Minor points

(3) Abstract:

“we propose a robust approach that identifies the strong inverse relationship for certain biomarkers”

Consider the following alternative:

“we propose a robust approach that identifies a strong inverse relationship of lifespan with certain biomarkers”

Response: We agree that this is a better way of putting it. We have implemented it as suggested.

(4) Page 4:

“our mathematical formalist predicts an inverse relationship between Adj.AROCM and Lifespan”

Repeated statement.

Response: we have corrected it.

(5) Figure S14:

Panels a and b are not explicitly referenced in the caption. Second, what are the vertical red bars in panels a and b showing?

Response: In response, we have expanded the caption. The vertical red bars are the mean values across all species.

(6) Discussion:

“These findings collectively underscore the significance of selecting an appropriate chromatin context for defining the AROCM.”

This wording would be appropriate, if the goal was to predict lifespan. In that case one would want to choose CpGs that are most strongly correlated with lifespan. However, the spin of this publication is more mechanistic: it directly addresses the process of DNA methylation. Thus, analyzing e.g. TxEx4 is not 'inappropriate'. Instead, that analysis reveals that those CpGs seem to be under different regulatory influence. Hence, I would prefer the following wording instead:

"These findings collectively underscore that the chromatin context substantially affects the relationship between AROCM and 1/Lifespan."

Response: We agree that this is a far better way of putting it. We have implemented it as suggested.

Detailed response to reviewer #2

In response to the reviewer's comments, the authors have removed their unsubstantiated claims and corrected the numerous errors in the formulas. The major claim that methylation negatively scales with maximum lifespan has been already published by

Crofts et al. in Nature Aging 2024. As outlined in the original reviewer comments, this paper was preprinted already earlier and is still ignored by the authors with the argument that Dr. Horvath described the idea in a TEDx video. It is unscholarly to refuse credence to authors who published the scientific paper describing the finding. Taken together, the advance provided by the current manuscript in comparison to the state of the art is unclear. The listing of the mathematical formulas could be useful to bioinformatics specialists in the methylation clock field.

Response: We now cite the paper by Crofts et al. published in Nature Aging 2024 in the Introduction and elsewhere. To the best of our knowledge, we are citing all relevant articles. Our article significantly surpasses the basic premise that methylation rate of change has an inverse relationship with maximum lifespan. We have discovered that this correlation is considerably modulated by the chromatin states.

Important original contributions of our work include.

1) We establish that the rate of methylation change and its correlation with lifespan are intricately linked to chromatin state.

2) We conduct a comprehensive analysis of the rate of change throughout the entire lifespan, detailing life course equations for clear reference.

3) We rigorously derive a proportionality law that aligns the rate of change in young animals with that observed in older counterparts.

4) We introduce innovative methods for addressing biased ascertainment issues, presenting concepts such as the adjusted rate of change and adjusted correlation.

5) We clarify that the correlation of age is not directly associated with lifespan.

Importantly, the mathematical formulas we present offer more than a mere enumeration; they provide a sophisticated analytical framework for researchers investigating the dynamics of change rates as they pertain to maximum lifespan.

We trust that these clarifications and additions underscore the scholarly merit and novelty of our manuscript.

Additional specific comments:

• **The article is still poorly written. Some key points are written in a convoluted way or point to the middle of the methods section without more explanation. While the main text doesn't need to contain all the mathematical details it should guide the reader more clearly and make the advance understandable without having to read the whole methods section.**

Response: We have changed the presentation. In particular, we have made more detailed references to the Methods section by specifying the subsections and equations relevant to the stated results.

• **One of the main aim is to show that it is beneficial to correct for imperfect sample biases. This is interesting to note and indeed a lot of people in the field might not directly be aware of this. How would the (un-)adjusted AROCM vs. log lifespan look like if only considering strata with roughly equal SDs? If the adjustment is working without introducing artificial biases the results should not change.**

Response: As suggested, the result do not change. In response to the recommendation, Supplementary figure S5 has been added to closely replicate the insights of main Figure 5, elucidating the relationship between AROCM and lifespan across various strata

characterized by similar SD(R) values. We have categorized these strata by the range of SD(R) values, specifically into three distinct strata: (0, 0.2), (0.2, 0.4), and (0.4, 0.6), to facilitate a stratified analysis.

The following was added to subsection “Adjusted AROCM approximates the inverse of maximum lifespan” in the Results:

Our findings underscore the efficacy of the adjusted AROCM in addressing the issue of strata with widely varying values of $\text{sd}(R)$. Traditionally, one might handle the variability in $\text{sd}(R)$ by limiting the analysis to strata that have approximately similar $\text{sd}(R)$ values. To demonstrate that our principal findings are consistent using this conventional method, we conducted the same analysis on strata with comparable $\text{sd}(R)$ values. The results were compelling: we observed strong correlations between AROCM and $1/\text{Lifespan}$, with correlation coefficients of $r = 0.99$ for strata within $\text{sd}(R) \in (0.2, 0.4)$, and $r = 0.97$ for strata within $\text{sd}(R) \in (0.4, 0.6)$, respectively (see Supplementary Figure S5). Furthermore, a stratified analysis reaffirmed our key findings from the dog breed study (refer to Supplementary Figure S6).

Reviewer 1 does have a valid point on the circular reasoning, so the statement of overfitting in the discussion is indeed important.

Response: As pointed out by the reviewer, this was already addressed in our previous response to reviewers. In particular, please see the following paragraph in the discussion:

However, these adjustments are not without issues, including the risk of overfitting and challenges in interpretation. We recommend presenting both the unadjusted and adjusted rate of change estimates in findings, as done in our research. Our results remain consistent whether using adjusted or unadjusted estimates.

Reviewers' Comments:

Reviewer #1:

Remarks to the Author:

First of all I would like to apologize to the authors for the long delay in my review. I only have two very small remaining points.

p. 4: "Strong negative correlations between lifespan can be further ..."
"correlation *with* lifespan"

p. 5: "To demonstrate that our principal findings are consistent using this conventional method, we conducted the same analysis on strata with comparable $\lambda_{sd}(R)$ values. The results were compelling: we observed strong correlations between AROCM ..."
Use a scientifically more neutral (less emotional) language, such as "The results confirmed our earlier findings: we observed ..."

Reviewer #2:

Remarks to the Author:

The authors have responded to the remaining comments adequately. They improved the writing and presentation of the manuscript so that it is now more accessible and provides a useful source for the aging clock field. The sentence "A promising avenue for research is examining the disparities in methylation rate changes between genders." needs to refer to sex not to gender.

Detailed response to reviewer #1

Reviewer: First of all I would like to apologize to the authors for the long delay in my review.

I only have two very small remaining points.

Response: No problem at all. Thank you again for your very helpful comments and careful reading.

**p. 4: “Strong negative correlations between lifespan can be further ...”
“correlation *with* lifespan”**

Response: Done.

p. 5: “To demonstrate that our principal findings are consistent using this conventional method, we conducted the same analysis on strata with comparable $\text{sd}(R)$ values. The results were compelling: we observed strong correlations between AROCM ...”

Use a scientifically more neutral (less emotional) language, such as “The results confirmed our earlier findings: we observed ...”

Response: We have changed the sentence to “The results confirmed our earlier findings of strong correlations between \$AROCM\$ and \$1/\text{Lifespan}\$, ...”

Detailed response to reviewer #2.

The authors have responded to the remaining comments adequately. They improved the writing and presentation of the manuscript so that it is now more accessible and provides a useful source for the aging clock field. The sentence “A promising avenue for research is examining the disparities in methylation rate changes between genders.” needs to refer to sex not to gender.

Response: Thank you for the careful reading. We have implemented the requested change.